# A Differentiable Semantic Metric Approximation in Probabilistic Embedding for Cross-Modal Retrieval

**Hao Li[1]**
18th.leolee@gmail.com

**Jingkuan Song[1]\***
jingkuan.song@gmail.com

**Lianli Gao[1]**
lianli.gao@uestc.edu.cn

**Pengpeng Zeng[1]**
is.pengpengzeng@gmail.com

**Haonan Zhang[1]**
zchiowal@gmail.com

**Gongfu Li[2]**
gongfuli@tencent.com

[1]Center for Future Media, University of Electronic Science and Technology of China
[2]Tencent Wechat Group, Shenzhen, China.

## Abstract

Cross-modal retrieval aims to build correspondence between multiple modalities by learning a common representation space. Typically, an image can match multiple texts semantically and vice versa, which significantly increases the difficulty of this task. To address this problem, probabilistic embedding is proposed to quantify these many-to-many relationships. However, existing datasets (*e.g.*, MS-COCO) and metrics (*e.g.*, Recall@K) cannot fully represent these diversity correspondences due to non-exhaustive annotations. Based on this observation, we utilize semantic correlation computed by CIDEr to find the potential correspondences. Then we present an effective metric, named Average Semantic Precision (ASP), which can measure the ranking precision of semantic correlation for retrieval sets. Additionally, we introduce a novel and concise objective, coined Differentiable ASP Approximation (DAA). Concretely, DAA can optimize ASP directly by making the ranking function of ASP differentiable through a sigmoid function. To verify the effectiveness of our approach, extensive experiments are conducted on MS-COCO, CUB Captions, and Flickr30K, which are commonly used in cross-modal retrieval. The results show that our approach obtains superior performance over the state-of-the-art approaches on all metrics. The code and trained models are released at https://github.com/leolee99/2022-NeurIPS-DAA.

## 1 Introduction

Vision-language approaches have demonstrated success in various multimodal tasks, such as video question answering [57], image captioning [56], and text-to-image generation [51]. As one of the most fundamental multimodal tasks, image-text retrieval is in great demand due to the explosive increase of multimedia data in social activities. Given an image or a text as a query, it aims to retrieve the most relevant items in the database of other modalities. Although significant progress has been made [21, 12, 43, 37, 55, 26, 25], this task remains challenging since the heterogeneity across different modalities. To tackle this problem, most existing methods map the query and retrieval items from distinct modalities into a common embedding space, which allows computing the similarity between different modalities.

Deep neural networks have achieved great success in various applications [58, 53]. In cross-modal retrieval, a line of methods [21, 39, 12] extracts representations of images and texts by constructing a

---

*Corresponding author.

36th Conference on Neural Information Processing Systems (NeurIPS 2022).

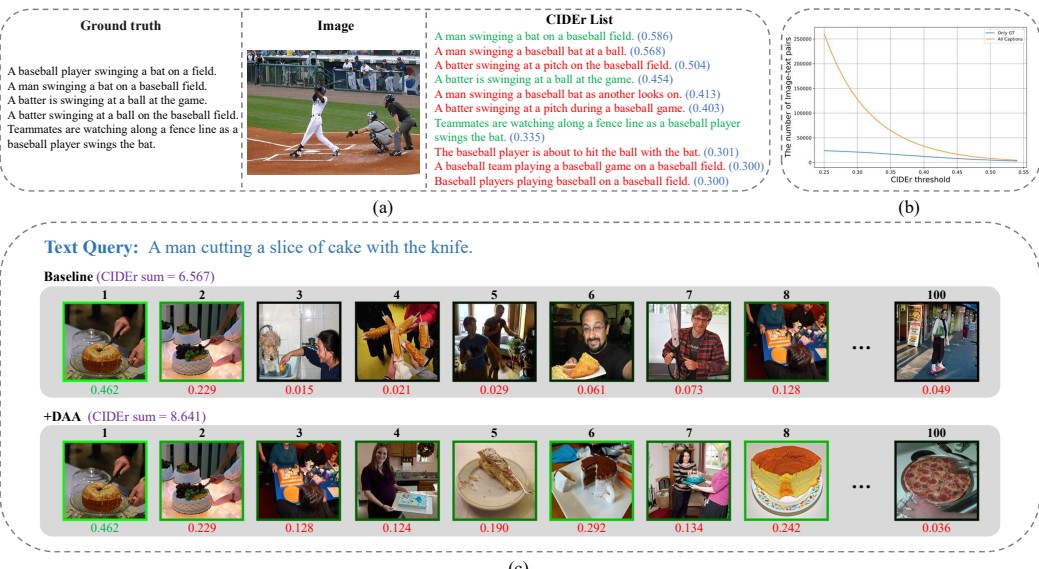

Figure 1: (a) An example of the image and its five ground truth captions. The CIDEr list gives 10 captions, followed by their CIDEr scores in the bracket with ground truth marked green, which are computed with all ground truth shown on the left. The captions with a CIDEr score greater than 0.3 can almost describe the image correctly (the last three captions in the CIDEr list). (b) The number of image-text pairs greater than the CIDEr threshold. There are a large number of image-text pairs with high semantic correspondence besides the ground truth. The number of potential positive pairs is about 4 times the number of annotated positive pairs, while CIDEr score is greater than 0.3. (c) The top 100 items of two retrieved sets predicted by two approaches. Below each image is the semantic similarity computed by CIDEr. Both retrieved sets obtain the same score by AP or Recall@K (only the first retrieved item is in ground truth). Nonetheless, compared with the first retrieved set, the second one predicted by the model using DAA is more semantically related to the text query. It shows that the similarities computed by CIDEr are generally consistent with the real semantic similarities between the text queries and the images.

two-stream deep neural network to learn the correspondence of inter-modalities well. Another thread of work [43, 37, 54] focuses on the relationship within the intra-modality. Recent studies [26, 25, 29] have shown that vision-language pre-training is effective to learn generic representations from massive image-text pairs. These methods have achieved impressive results on existing benchmarks. However, most of the prior works [21, 39, 12, 43, 10, 37, 54] use projection functions to map a sample as a deterministic embedding, which does not definitely account for the potential many-to-many relationships in the existing datasets (*e.g.*, MS-COCO) with non-exhaustive annotations. Concretely, as shown in the second row of Figure 1 (c), given a query caption such as "A man cutting a slice of cake with the knife," only one annotated image (the first image) is considered as positive, whereas some other images are also correct at the semantic level. To tackle this problem, Chun *et al*. [7] introduce probabilistic embedding for cross-modal retrieval, which is powerful to represent many-to-many relationships, by converting the samples into various representations according to a distribution.

Although probabilistic models can successfully capture potential many-to-many relationships, the diversity of probabilistic embedding heavily depends on the well-matched datasets, which are expensive to annotate manually. Furthermore, commonly used datasets for the cross-modal retrieval task suffer from non-exhaustive annotations, which miss several positive pairs. These missing positive pairs (MPPs) enormously limit the multiplicity expressed in training and evaluation processes. To excavate these MPPs, we propose a semantics-based many-to-many correspondence mining strategy, which represents the relevance between images and captions through semantic similarity. Specifically, the semantic metric CIDEr [45] is used to calculate semantic similarity between images and captions, *i.e.*, all image-text pairs with CIDEr scores greater than the threshold are regarded as positives. Then, we observe that a caption with a CIDEr score greater than 0.3 can typically describe images correctly (see Figure 1 (a)). To figure out the data scale of MPPs in commonly used datasets, the number of potential positives with different CIDEr thresholds in the MS-COCO 5K test set is measured in

Figure 1 (b). It can be seen that 128,050 image-text pairs with CIDEr scores greater than 0.3, which means that each image equally matches 25.61 captions (only 5 annotated captions) on the semantic. The results powerfully demonstrate a large number of potential positives are wrongly regarded as negatives, severely impeding diversity in training and evaluation processes.

Nonetheless, most optimization methods [12, 15, 13, 47] and metrics revolve around the accurate prediction of positives in the ground truth, ignoring the correlations between other retrieved items and the query. As the situation in Figure 1 (c), the retrieved set at the bottom is more in line with our expectations than the one above, even though they obtain the same score by the metrics only focusing on the ground truth, such as AP or Recall@K. It means that ground-truth-based metrics are hard to fully represent the multiplicity for cross-modal retrieval. To address these problems, we carry out the following work.

In this work, we give a solution to alleviate the hindrance of MPPs to multiplicity for probabilistic representations in the training process. Firstly, to evaluate with non-ground-truth relevant items, we propose a new evaluation metric, named Average Semantic Precision (ASP). ASP computes the average deviation between prediction ranking and semantic ranking. It indicates the semantic ranking precision of each retrieved item. We utilize CIDEr as the criterion for judging the semantic correlations of captions. Secondly, we propose a concise and novel Differentiable ASP Approximation (DAA) method, which optimizes ASP directly by relaxing the discrete ranking function in the non-differentiable ASP with a sigmoid function. Eventually, our approach achieves state-of-the-art performance on several multiplicity metrics.

The main contributions and novelties of this paper can be summarized below. i) We count the number of image-text pairs with strong semantic correlations on the most widely used dataset MS-COCO. The result shows incompletely annotating MPPs is a crucial factor restricting many-to-many mapping for probabilistic embedding. Thus, we propose an effective evaluation metric named ASP instead of the existing ones, which are powerless to reflect the correlations between each retrieved item and the query. ii) To eliminate the adverse effect of MPPs in the training process, we propose a novel and concise metric learning method DAA, which approximates and optimizes ASP directly. iii) To verify the effectiveness of our method, we conduct experiments on two challenging benchmarks and multiple sensible evaluation metrics.

## 2  Related Work

In this section, we briefly introduce some recent developments in cross-modal retrieval and metric learning.

### 2.1  Cross-modal Retrieval

Cross-modal retrieval requires a coordinated representation [3] that allows computing the similarity between the query and the retrieved items from distinct modalities. Early works [15] adopt canonical correlation analysis (CCA) to learn a linear projection function and maximize the correspondences between positive images and texts. Faghri *et al*. [12] first uses a triplet loss, which only samples the hardest negative to boost the matching performance. Nonetheless, one of the drawbacks of these approaches is their neglect to capture the correspondences between two modalities at a fine-grained level. Some prior works [23, 24, 10] capture fine-grained local semantic alignments between visual regions and words. SCAN [23] proposes learning to extract region features by an object detector pre-trained on Visual Genomes [22]. VSRN [24] adopts a graph neural network for semantic reasoning on the region proposals generated by a Faster-RCNN [36] detector. It gains significant performance improvements with the large increase in computing expenses. SGRAF [10] constructs a similarity graph to identify complex matching patterns and adopts a similarity attention filtration module to eliminate the interferences of no-meaningful alignments. However, the aforementioned approaches ignore the potential many-to-many correspondences in the datasets with non-exhaustive annotations. Parekh *et al*. [32] construct an extended MS-COCO dataset, *i.e*., CrissCrossed Caption (CxC) dataset, which provides denser annotations to excavate more potential positive relationships between unassociated images and captions. Recently, Chun *et al*. [7] use probabilistic embedding in cross-modal retrieval to obtain many-to-many alignments between images and captions. A transition of R-Precision incorporating plausible matches by class-based similarity is adopted to calculate the multiplicity score. Additionally, some works [59, 42] explore the realistic distribution to

make the probabilistic models express more accurate many-to-many-relationships. Although some progress [32, 41, 7] has been made in capturing potential correspondences, MPPs still seriously impede the multiplicity performance of cross-modal models. To tackle this problem, we employ semantic metrics on representing the semantic correlations between images and captions to excavate the potential positive pairs and evaluate the multiplicity of cross-modal models.

## 2.2 Metric Learning

Metric Learning maps input data into an embedding space so that distance metrics can be applied easily. A great body of work [2, 1, 48, 44, 40, 6, 49] has focused on metric learning. For instance, contrastive [6] loss and triplet [49] loss are two of the most widely used methods. The former forces all positive instances closer, while negative pairs are only separated by a fixed margin. In the latter, positive pairs are pushed closer than negative pairs. Although impressive success has been achieved, the motivation to only minimize distance is limited. The aforementioned methods ignore the ranking orders, which are essential for rank-based metrics such as Average Precision (AP). Brown *et al.* [5] propose a differentiable AP approximation loss that can optimize AP directly in image retrieval. Compared with triplet loss, it shows better performance in AP, a ranking-based metric. Impressive progress [44, 40, 11, 18, 17, 33, 38] in optimizing AP has been achieved, but AP is a limited metric, which only focuses on the ground truth. In our work, a ranking-based metric ASP is proposed to exploit potential semantic information. Besides, we propose a Differentiable ASP Approximation (DAA) to optimize ASP directly. To our knowledge, DAA is the first work that uses direct metric optimization for cross-modal retrieval.

## 3  Preliminaries

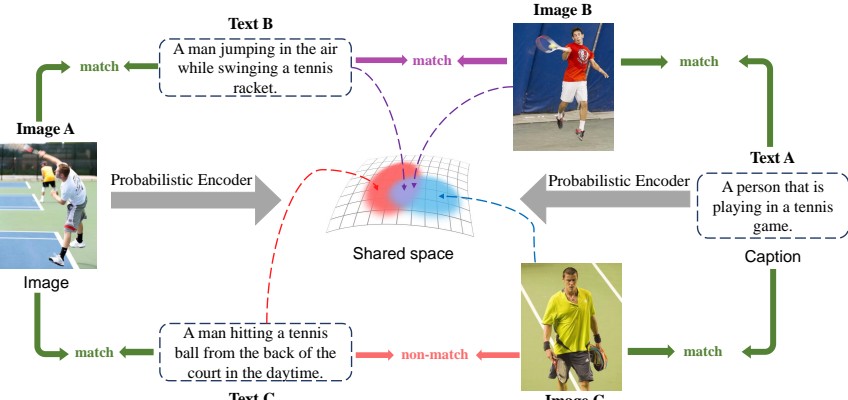

Figure 2: An example of probabilistic embedding to represent multiplicity. For each positive pair (*e.g.*, image A and text A), it will generate two distributions through a probabilistic encoder. For the distribution coming from image A (marked red in shared space), captions within the distribution (*e.g.*, Text B and C) all match image A. Similarly, Image B and C can match Text A. The image and text on the intersection of two distributions (marked purple in shared space) will match each other (*e.g.*, Image B and Text B), while non-match on the outside.

**Probabilistic embedding.** Chun *et al.* [7] have a previous attempt at Probabilistic Cross-Modal Embedding (PCME) to address the multiplicity of cross-modal correlation. PCME models each sample as a normal distribution (See Figure 2), which could be described by mean and variance together. Therefore, the key to generating probabilistic embedding is determining the mean embedding and the variance matrix. Given an image $v \in \mathcal{V}$ and a caption $t \in \mathcal{T}$ as inputs, probabilistic model will learn their normal distributions as:

$$p(E(v)|v) \sim N(h_{\mathcal{V}}^{\mu}(f_v), h_{\mathcal{V}}^{\sigma}(\text{diag}(f_v)))$$
$$p(E(t)|t) \sim N(h_{\mathcal{T}}^{\mu}(f_t), h_{\mathcal{T}}^{\sigma}(\text{diag}(f_t)))$$

(1)

where $f_v$ is the feature map extracted from the visual encoder and $f_t$ is the feature sequence extracted from the textual encoder. $E(\cdot)$ is a function mapping input to probabilistic embedding. $h^{\mu}$ and $h^{\sigma}$

are two head modules used to compute the mean embedding and the variance matrix in PCME [7], respectively. To learn the joint probabilistic embedding, the soft cross-modal contrastive loss is used [14], which can be defined as:

$$\mathcal{L}_{v,t}(\theta) = \begin{cases} -\log p_\theta(m|v,t) & \textit{if } v,t \textit{ is a match} \\ -\log(1 - p_\theta(m|v,t)) & \textit{otherwise} \end{cases}, \tag{2}$$

where $\theta = (\theta_V, \theta_T)$ are the parameters for the visual and the textual encoders, respectively. Given an image-text pair $(v, t)$ as input, $p_\theta(m|v,t)$ is the match probability from Euclidean distances can be similarly denoted as:

$$p_\theta(m|v,t) \approx \frac{1}{J^2} \Sigma_j^J \Sigma_{j'}^J \phi(-a||E(v)^j - E(t)^{j'}||_2 + b), \tag{3}$$

where $E(v)^j$ and $E(t)^{j'}$ are samples from the embedding distribution $p(E(v)|v)$ and $p(E(t)|t)$ via a Monte-Carlo estimator [31], respectively. $(a, b)$ are learnable scalars and $\phi(\cdot)$ is a sigmoid function.

# 4 Approach

In this section, we elaborate on the proposed method. In Section 4.1, we first delineate our novel evaluation metric, *i.e.*, Average Semantic Precision (ASP), which can calculate the semantic correlation scores of the retrieved set easily and precisely, even in noisy datasets. In Section 4.2, we detail how to optimize ASP directly in probabilistic embedding by using Differentiable ASP Approximation (DAA).

## 4.1 Average Semantic Precision (ASP)

Existing diversity metrics [7, 32] for not well-aligned datasets are hard to fully represent the semantic ranking accuracy of retrieved sets limited by incomplete correlation labels, such as PMRP [7] or CxC [32]. The former treats samples in MS-COCO [27] whose class label difference numbers are less than 2 as positives, but it will make a lot of false matches. The latter manually annotates the testing set of MS-COCO, but still misses many potential positive pairs. Meanwhile, both of them are merely applicable to MS-COCO, *i.e.*, they are not universal. To solve these problems, we propose Average Semantic Precision (ASP). Given a query set $\Psi = \{Q_i, i = 0, ..., n\}$, a query $q \in \Psi$ can simply calculate the semantic correlation scores of the retrieved set $\Omega = \{G_i, i = 0, ..., m\}$ in various image-text retrieval datasets. Mathematically,

$$ASP_q = \frac{1}{|\Omega|} \sum_{i \in \Omega} \frac{\mathcal{M}in(\mathcal{R}(i, C), \mathcal{R}(i, S))}{\mathcal{M}ax(\mathcal{R}(i, C), \mathcal{R}(i, S))}, \tag{4}$$

where $\mathcal{R}(i, C)$ and $\mathcal{R}(i, S)$ refer to the rankings of the instance $i$ in semantic list $C$ (defined as Eq. 6) computed by CIDEr [45], and similarity list $S$ (defined as Eq. 5) predicted by the model. In Eq. 4, ASP utilizes the overall gap between the semantic ranking and the similarity ranking to represent the precision of a retrieved set.

$$S = \{s_i = \left\langle \frac{E(q)}{\|E(q)\|} \cdot \frac{E(i)}{\|E(i)\|} \right\rangle, \quad i = 0, 1, ..., m\}, \tag{5}$$

$$C = \{c_i = SEM(i, q), \quad i = 0, 1, ..., m\}, \tag{6}$$

The $SEM(i, q)$ is the semantic correlation score between query $q$ and retrieved item $i$. When $q$ and $i$ refer to an image and a text, we will select all positive texts (5 texts in MS-COCO) corresponding to the image, and calculate their semantic similarities with $i$ through the CIDEr metric, respectively. Eventually, the mean of these semantic similarities will represent the semantic similarity between $q$ and $i$. Conversely, the positive texts corresponding to image $i$ will be selected to calculate the semantic correlation scores with text $q$. The specific equation is as follows:

$$SEM(i, q) = \begin{cases} \dfrac{1}{P} \sum_{j=1}^{P} \psi(i, \mathcal{P}os(q)_j), & \textit{if } q \textit{ is an image} \\ \dfrac{1}{P} \sum_{j=1}^{P} \psi(\mathcal{P}os(i)_j, q), & \textit{if } q \textit{ is a text} \end{cases} \tag{7}$$

where $\mathcal{P}os(q)$ is the positive texts of $q$, $P$ is the number of texts in $\mathcal{P}os(q)$, $\psi$ stands for the CIDEr metric.

## 4.2 Differentiable ASP Approximation (DAA)

Contrastive [6] and triplet [49] losses are two widely used metric learning methods, which are driven by minimizing distance. They both ignore the ranking order, which is important for ranking-based metrics. Thus, the ranking-based metrics will be indirectly optimized in these methods. ASP is a ranking-based metric that is neither differentiable nor decomposable. Consequently, we propose a Differentiable ASP Approximation (DAA) to optimize ASP directly.

Given the ranking-based metric ASP, the reason why it is non-differentiable is that the derivative of a ranking function $\mathcal{R}$ for instance $i$ defined in Eq. 8 is either gradient zero or discontinuous [34], resulting in the inability to be optimized with gradient-based approaches.

$$\mathcal{R}(i, U) = 1 + \sum_{j \in U, j \neq i} \mathbb{1}\{(u_i - u_j) < 0\}, \tag{8}$$

where $\mathbb{1}\{\cdot\}$ serves as an Indicator function, which select 1 while $(u_i - u_j) < 0$, otherwise select 0 [34]. $U$ is the CIDEr score set $C$ or the similarity score set $S$. The difference matrix of $D^U \in \mathbb{R}^{m \times m}$ can be computed as:

$$D^U = \begin{bmatrix} u_1 & \cdots & u_m \\ \vdots & \ddots & \vdots \\ u_1 & \cdots & u_m \end{bmatrix} - \begin{bmatrix} u_1 & \cdots & u_1 \\ \vdots & \ddots & \vdots \\ u_m & \cdots & u_m \end{bmatrix}, \tag{9}$$

The ASP for the query $q$ can be converted from Eq. 4 to:

$$ASP_q = \frac{1}{|\Omega|} \sum_{i \in \Omega} \frac{1 + \mathcal{M}in(\sum_{j \in \Omega, j \neq i} \mathbb{1}\{D_{ij}^C > 0\}, \sum_{j \in \Omega, j \neq i} \mathbb{1}\{D_{ij}^S > 0\})}{1 + \mathcal{M}ax(\sum_{j \in \Omega, j \neq i} \mathbb{1}\{D_{ij}^C > 0\}, \sum_{j \in \Omega, j \neq i} \mathbb{1}\{D_{ij}^S > 0\})}, \tag{10}$$

Inspired by [5], we utilize a sigmoid function $\phi(\cdot)$ instead of the Indicator function $\mathbb{1}\{\cdot\}$ to make ASP differentiable, which could achieve direct optimization of ASP. After substituting $\phi(\cdot)$ into Eq. 8, the approximation of ASP is denoted as:

$$ASP_q = \frac{1}{|\Omega|} \sum_{i \in \Omega} \frac{1 + \mathcal{M}in(\sum_{j \in \Omega} \phi(D_{ij}^C), \sum_{j \in \Omega} \phi(D_{ij}^S))}{1 + \mathcal{M}ax(\sum_{j \in \Omega} \phi(D_{ij}^C), \sum_{j \in \Omega} \phi(D_{ij}^S))}, \tag{11}$$

which is differentiable and could be optimized with gradient-based methods. Therefore, the final loss function is defined as:

$$\mathcal{L}_{ASP} = \frac{1}{|\Psi|} \sum_{k \in \Psi} (1 - ASP_k), \tag{12}$$

## 5 Experiment

In this section, we conduct experiments to verify the effectiveness of DAA, which could optimize ASP directly. We describe the evaluation protocol, including datasets and metrics, as well as the implementation details. Then we report experimental results on MS-COCO [27], CUB Captions [7], and Flickr30K [52] for the cross-modal retrieval task. After that, we carry out a series of ablation studies to verify the effectiveness of DAA in improving multiplicity. Meanwhile, we present an experimental analysis of the effectiveness of DAA.

### 5.1 Evaluation Protocol

#### 5.1.1 Datasets

We conduct experiments on MS-COCO [27] and CUB Captions [7] to show the performance of probabilistic embedding using DAA for cross-modal retrieval. Although our method is proposed to improve the multiplicity of the probabilistic model, it seems that the proposed model can be integrated into any cross-modal retrieval method. Thus, we also employ DAA in non-probabilistic embedding on MS-COCO and Flickr30K [52] to explore the generalization of our method.

**MS-COCO** is a widely used dataset for the cross-modal retrieval task. It contains 123,287 images with five captions per image. We follow the data partition in [20], which consists of 113,287 images

for training, 5,000 images for validation, and 5,000 images for testing. We report our results on both 5K and 1K testing sets (the average over 5-folds of 1K testing images).

**CUB Captions** [7, 35, 46] is a benchmark proposed to reduce the impact of false matching for probabilistic embedding in MS-COCO. It contains 11,788 images of 200 fine-grained bird categories with 10 captions per image. All the image-text pairs belonging to the same class are considered positive pairs, so false negatives rarely exist. Meanwhile, the homogeneity of image-text pairs in the same class could greatly suppress false positives. We follow the class splits in [50], which consists of 150 classes for training and validation, and the remaining 50 for testing.

**Flickr30K** [52] is a widely used dataset for the cross-modal retrieval task. It contains 31783 images with 5 captions per image. We follow the split in [13], which consists of 1000 images for validation, 1000 images for testing, and the rest for training.

### 5.1.2 Metrics

We take **Recall@K** (R@K), a widely used metric in cross-modal retrieval, as a measurement in both benchmarks. We report R@1, R@5, and R@10 for a comprehensive evaluation. However, R@K only focuses on the position of the first relevant item according to the ground truth, reflecting limited information.

**R-Precision** (R-P) is an alternative proposed by Musgrave *et al.* [30]. For each query, R-P computes the ratio of matched items in the top-$r$ retrieved items, where $r$ is the number of ground-truth matches. In short, for all matched items, the fewer negatives before, the higher the R-P score is. Compared with R@K, R-P could better evaluate the one-to-many matching for cross-modal retrieval. We use R-P to evaluate the performance of DAA on CUB Captions.

As MS-COCO, we take **Plausible Match R-Precision** (PMRP) [7] metric as a measurement, which utilizes the class labels for MS-COCO to find out all hidden positive image-text pairs. Concretely, PMRP regards the pair $(v, t)$, whose binary label vectors $y^v, y^t \in \{0, 1\}^{d_{label}}$, differ at most at $\zeta$ positions ($\zeta \in \{0, 1, 2\}$), as positive matches.

The above metrics can only reflect the precision of ground-truth items, ignoring the correlations between other non-matched retrieved items and queries. **Average Semantic Precision** (ASP) is a metric proposed to reflect the models' multiplicity by evaluating the retrieved sequence's semantic relevance. Therefore, we refer to ASP (Eq. 4) as an evaluation measurement for both benchmarks.

### 5.2 Implementation Details

For the probabilistic model, we use ResNet [16] and the basic version of the pre-trained BERT [8] as visual and textual encoders. There are three phases in the training proceeds: The first is a pre-training phase, where only the head modules in the visual encoder and all parameters in the textual encoder are trained. A warm-up phase lasting three epochs is used in the pre-training phase. Finally, all parameters will be trained in the fine-tuning phase. Concretely, for MS-COCO, we use ResNet-152 backbone with embedding dimension $D = 1024$. For CUB, we use ResNet-50 backbone with embedding dimension $D = 512$. Meanwhile, We utilize AdamP Optimizer [19] for model training with the batch size of 128 and 150 for MS-COCO and CUB, respectively. For both datasets, we set the probabilistic sample number as 5 and the initial learning rate as 0.00015 with the cosine learning rate scheduler [28]. Besides, models are always trained on a single RTX A6000 GPU with Cutout [9] and random caption dropping [4] augmentation strategies with 0.2 and 0.1 erasing ratios, respectively.

For the non-probabilistic model, we take the publicly released code of SGRAF [10] to employ our method. There are two independent modules in the SGRAF network (SGR and SAF). Consistent settings are used for the training of SGR and SAF as follows. We conduct all experiments on 4 RTX A6000 GPUs with the mini-batch size of 128. Besides, the Adam optimizer [19] is employed for model training with the initial learning rate of 0.0006 for the first 30 (40) epochs and decays it to 0.00006 for the last 10 epochs on MS-COCO [27] (Flickr30K [52]). In the evaluation stage, the performance of SGRAF is predicted by averaging the prediction similarities of SGR and SAF.

Table 1: Comparison on MS-COCO

| Method | 1K Test Images | | | | | | 5K Test Images | | | | | |
| | i2t | | | t2i | | | i2t | | | t2i | | |
| | PMRP | R@1 | ASP | PMRP | R@1 | ASP | PMRP | R@1 | ASP | PMRP | R@1 | ASP |
|---|---|---|---|---|---|---|---|---|---|---|---|---|
| VSE++ | - | 64.6 | - | - | 52.0 | - | - | 41.3 | - | - | 30.3 | - |
| PVSE K=1 | 40.3 | 66.7 | - | 41.8 | 53.5 | - | 29.3 | 41.7 | - | 30.1 | 30.6 | - |
| PVSE K=2 | 42.8 | 69.2 | - | 43.6 | 55.2 | - | 31.8 | 45.2 | - | 32.0 | 32.4 | - |
| PCME | 45.0 | 68.8 | 54.7 | 46.0 | 54.6 | 55.3 | 34.1 | 44.2 | 54.5 | 34.4 | 31.9 | 55.0 |
| PCME-bert | 44.3 | 69.2 | 54.4 | 45.3 | 56.2 | 54.4 | 33.0 | 45.6 | 54.2 | 33.4 | 33.2 | 54.2 |
| **+DAA** | **45.9** | **71.2** | **58.4** | **46.7** | **57.8** | **57.2** | **34.6** | **48.2** | **58.3** | **34.9** | **35.1** | **57.0** |
| SAF | 46.2 | 77.1 | 54.7 | 47.3 | 62.1 | 54.8 | 34.7 | 55.1 | 54.5 | 35.3 | 40.5 | 54.5 |
| **+DAA** | **47.1** | **78.0** | **67.2** | **48.7** | **62.8** | **61.6** | **35.7** | **56.2** | **67.0** | **36.6** | **40.5** | **61.4** |
| SGR | 44.0 | 77.2 | 53.9 | 46.0 | 61.9 | 54.6 | 32.7 | 55.8 | 53.7 | 34.2 | 40.1 | 54.4 |
| **+DAA** | **46.4** | **78.0** | **68.5** | **48.6** | **62.6** | **62.8** | **35.3** | **56.5** | **68.4** | **36.9** | **40.8** | **62.6** |
| SGRAF | 47.1 | 79.4 | 54.5 | 48.0 | 63.9 | 55.0 | 35.5 | 59.0 | 54.3 | 35.9 | 42.6 | 54.8 |
| **+DAA** | **48.1** | **80.2** | **68.3** | **49.6** | **65.0** | **62.7** | **36.6** | **60.0** | **68.2** | **37.5** | **43.5** | **62.5** |

## 5.3 Comparisions with State of The Arts

In this section, we compare our method with previous works using ResNet as the visual encoder on the two datasets. The comparison methods include VSE0 [12], VSE++ [12], PVSE [41], and PCME [7]. Besides, we utilize our method on 4 typically baselines in image-text retrieval including PCME-bert [7], SAF [10], SGR [10], and SGRAF [10]. PCME-bert is a variety of PCME using the basic version of BERT as the textual encoder. It achieves state-of-the-art performance in probabilistic approaches. For non-probabilistic approaches, we select three commonly used baselines with impressive performance, including SAF, SGR, and SGRAF.

**Results on MS-COCO.** Table 1 shows the quantitative results of MS-COCO 1K and 5K testing sets for image-to-text (i2t) and text-to-image (t2i) retrieval. We observe that our approach obtains a consistent improvement in performance on all metrics of each baseline. Exactly, PCME-bert using implementation strategy in Section 5.2 emerges better performance than PCME on R@1. Then, DAA is employed in the training process of PCME-bert, resulting in significant boosts on R@1, PMRP, and ASP. As for MS-COCO (1K), compared with PCME, we achieve absolute boosts on PMRP, R@1, and ASP in both i2t and t2i, which are 0.9%, 2.4%, 3.7% and 0.7%, 3.2%, 1.9%, respectively. As for MS-COCO (5K), compared with PCME, the boosts on PMRP, R@1 and ASP are 0.5%, 4.0%, 3.8% and 0.5%, 3.2%, 2.0%. Besides, for non-probabilistic approaches, SGRAF using DAA achieves state-of-the-art performance with R@1 of 80.2% (60.0%) and 65.0% (43.5%) for text and image retrieval in MS-COCO 1K (5K), separately. Meanwhile, the impressive improvement in PMRP and ASP means that SGRAF has achieved higher multiplicity performance even than the probabilistic model. The above results show that DAA could effectively enhance the scores of multiple metrics on various cross-modal models, which indicates that semantics is useful to boost models' performance in cross-modal retrieval.

Table 2: Comparison on CUB Captions

| Method | Image-to-text | | | Text-to-image | | |
| | R-P | R@1 | ASP | R-P | R@1 | ASP |
|---|---|---|---|---|---|---|
| VSE0 | 22.4 | 44.2 | - | 22.6 | 32.7 | - |
| PVSE K=1 | 22.3 | 40.9 | - | 20.5 | 31.7 | - |
| PVSE K=2 | 19.7 | 47.3 | - | 21.2 | 28.0 | - |
| PCME | 26.3 | 46.9 | 53.1 | 26.8 | 35.2 | 56.8 |
| PCME-bert | 27.2 | 49.2 | 52.9 | 27.8 | 36.3 | 56.0 |
| **+DAA** | **28.2** | **53.2** | **53.4** | **28.5** | **37.7** | **57.6** |

**Results on CUB Captions.** To demonstrate the robustness of the proposed approach for probabilistic models, we conduct experiments on CUB Captions proposed to evaluate the multiplicity of probabilistic models. Compared with MS-COCO, R-P is more suitable for CUB due to rare false pairs. The results on CUB are presented in Table 2. It can be seen that PCME-bert using DAA outperforms all the state-of-the-art methods. Especially for image-to-text retrieval, using DAA surpasses the

previous best method on R-P, R@1, and ASP with an increase of 1.9%, 6.3%, and 0.3%, respectively. Besides, for text-to-image retrieval, the model using DAA also outperforms the best counterpart by a significant margin (1.7% for R-P, 2.5% for R@1, and 0.8% for ASP).

**Results on Flickr30K.** To further demonstrate the robustness of DAA for non-probabilistic models, we employ SAF, SGR, and SGRAF trained with the proposed method DAA on Flickr30K. The results are shown in Table 3. We can observe that DAA still improves the performance of all models on both R@1 and ASP. Additionally, SGRAF trained with DAA achieves state-of-the-art performance with a 0.9% increase for R@1 and a 16.4% increase for ASP on i2t, as well as 1.4% for R@1 and 8.6% for ASP on t2i. The above results prove that DAA has a strong generalization for improving the performance of cross-modal models and demonstrates the effectiveness and importance of exploiting semantic information in cross-modal retrieval.

## 5.4 Ablation Studies

In this section, we perform several ablation studies on MS-COCO 5K to explore the superiority of DAA over the optimization method only focusing on ground truth (*e.g.*, Smooth-AP [5]), on the probabilistic model.

To explore the effect of DAA and Smooth-AP, we test the impact of using these two methods on multiple metrics. The results are shown in Table 4. First, we evaluate PCME-bert as the baseline, where the sample number is 5. Subsequently, we train our model with Smooth-AP and find that there are only a few improvements on PMRP, R@1, and ASP. Finally, we employ the DAA to the baseline, resulting in considerable boosts on all metrics, especially R@1 (2.6%

Table 3: Comparison on Flickr30K

| Method | i2t | | t2i | |
|---|---|---|---|---|
| | R@1 | ASP | R@1 | ASP |
| SAF | 73.8 | 49.0 | 56.7 | 50.0 |
| **+DAA** | **73.9** | **65.0** | **56.9** | **58.4** |
| SGR | 73.7 | 54.6 | 55.8 | 55.1 |
| **+DAA** | **73.8** | **65.5** | **56.6** | **59.0** |
| SGRAF | 77.1 | 49.4 | 58.5 | 50.6 |
| **+DAA** | **78.0** | **65.8** | **59.9** | **59.2** |

for i2t, 1.9% for t2i) and ASP (4.1% for i2t, 2.8% for t2i). The significant improvement of ASP shows that DAA shifts retrieved items with high relevance forward overall. These results implicate that excavating semantic correlation can not only improve the multiplicity of probabilistic models but also improve the traditional ground-truth-based metrics, such as R@K. The visualization of top 100 retrieved results is shown in Figure 1 (c).

The results in Table 4 prove that DAA can increase the semantic similarity in the whole retrieval set, which can be regarded as a set with large CIDEr score variance. To validate the effect of DAA for subsets with different CIDEr score variance, we rank the retrieved set according to the CIDEr score in descending order and sample the top N to compute ASP. We draw the contrastive curves of ASP in Baseline, DAA, and Smooth-AP [5] with N from 0 to the length of the whole retrieval set in Figure 3. We can observe that DAA always performs better than others, and the gap becomes larger along with the increase of N. The results indicate that DAA could enhance diversity performance in any retrieval set. Meanwhile, the curves of Smooth-AP and baseline are almost consistent, which shows that only focusing on ground truth is limited to improving multiplicity.

Table 4: Effect of Smooth-AP and DAA on MS-COCO 5K

| Method | Image-to-text | | | Text-to-image | | |
|---|---|---|---|---|---|---|
| | PMRP | R@1 | ASP | PMRP | R@1 | ASP |
| Baseline (B) | 33.0 | 45.6 | 54.2 | 33.4 | 33.2 | 54.2 |
| B+Smooth-AP | 33.0 | 45.4 | 54.2 | 33.5 | 33.3 | 54.2 |
| B+DAA | **34.6** | **48.2** | **58.3** | **34.9** | **35.1** | **57.0** |

To make clear the impacts of different methods on multiplicity in the training process, we test ASP in each epoch for baseline, Smooth-AP, and DAA. The results are shown in Figure 4, which provides a quantitative analysis of ASP against epochs. We can observe that with the increase of training rounds, the curve of the training strategy using DAA grows constantly. It substantially verifies that DAA effectively captures complex semantic correspondences and enhances multiplicity in the training process. The ASP curves of baseline and Smooth-AP strategies are almost invariant, while R@K

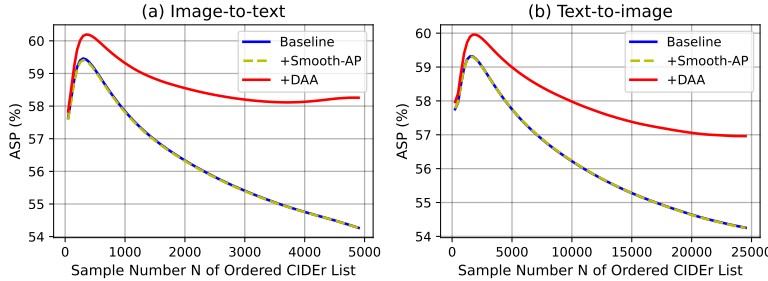

Figure 3: Ablation studies of the retrieval subsets with different CIDEr score variance on ASP metric. Given a retrieval set in descending order according to CIDEr score, we compute the ASP of top-N items. The variance will increase with the increase of N.

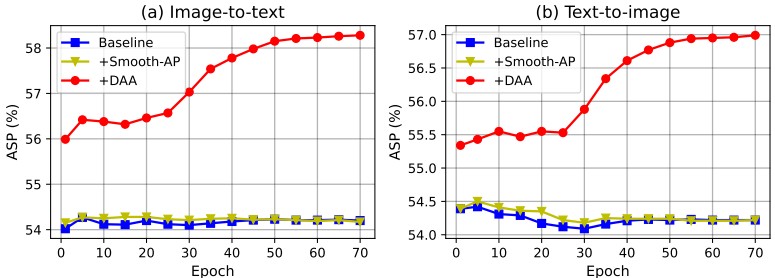

Figure 4: Ablation studies of different training strategies during training on ASP metric. The training process consists of 70 epochs. The first 3 epochs are the warm-up phase, the next 27 epochs are the pre-training phase, and the last 40 epochs are the fine-tuning phase.

improves constantly. This means that boosting the prediction accuracy of the items in ground truth may not improve multiplicity. All of these ablation studies strongly demonstrate that DAA is superior to the optimization method focusing only on ground truth in improving multiplicity.

## 6 Conclusion

This paper attempts to address a critical problem in probabilistic embedding for cross-modal retrieval, *i.e.*, the not well-aligned datasets and existing metrics can hardly train and evaluate the multiplicity performance of the model. To solve this problem, we introduce Average Semantic Precision (ASP) to evaluate the performance of multiplicity in these not well-aligned datasets. Moreover, we propose a novel Differentiable ASP Approximation (DAA), which could make the model diversified even if training in these noisy datasets. We have demonstrated the effectiveness of DAA for boosting the multiplicity of probabilistic embedding for cross-modal retrieval by outperforming the state-of-the-art models.

**Acknowledgements**

This study is supported by grants from National Key R&D Program of China (2022YFC2009903/2022YFC2009900), the National Natural Science Foundation of China (Grant No. 62122018, No. 62020106008, No. 61772116, No. 61872064).

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
