# Appendix: A Differentiable Semantic Metric Approximation in Probabilistic Embedding for Cross-Modal Retrieval

**Hao Li**[1]
18th.leolee@gmail.com

**Jingkuan Song**[1]*
jingkuan.song@gmail.com

**Lianli Gao**[1]
lianli.gao@uestc.edu.cn

**Pengpeng Zeng**[1]
is.pengpengzeng@gmail.com

**Haonan Zhang**[1]
zchiowal@gmail.com

**Gongfu Li**[2]
gongfuli@tencent.com

[1]Center for Future Media, University of Electronic Science and Technology of China
[2]Tencent Wechat Group, Shenzhen, China.

In this supplementary material, we discuss the following topics: Firstly, we discuss why we adopt Eq. 1 as the formulation of ASP in Appendix. A. Then, we analyze the differences between two multiplicity metrics, PMRP [1] and ASP, and give their advantages and shortcomings in Appendix. B, respectively. Furthermore, the effect of different semantic metrics on DAA is explored in Appendix. C. To figure out the applicable conditions of DAA, we use DAA in model training with different scale datasets (Appendix. D).

## A    How ASP Formulation is Designed?

The formulation of ASP in the paper is as Eq. 1. We design the formulation of ASP with reference to AP (Average Precision, Eq. 2).

$$ASP_q = \frac{1}{|\Omega|} \sum_{i \in \Omega} \frac{\mathcal{M}in(\mathcal{R}(i,C), \mathcal{R}(i,S))}{\mathcal{M}ax(\mathcal{R}(i,C), \mathcal{R}(i,S))}, \tag{1}$$

$$AP_q = \frac{1}{|S_P|} \sum_{i \in \omega} \frac{\mathcal{R}(i, S_P)}{\mathcal{R}(i, S_\omega)} \tag{2}$$

where $S_P = \{s_\zeta, \zeta \in \mathcal{P}_q\}, S_N = \{s_\xi, \xi \in \mathcal{N}_q\}$ are the positive and negative relevance score sets, respectively. Addtionally, $S_\omega = S_P \cup S_N$.

Intuitively, $\frac{\mathcal{R}(i,S_P)}{\mathcal{R}(i,S_\omega)}$ is the distance between the ranking of $i$-th item in $S_P$ and the ranking of it in $S_\omega$. Meanwhile, in $AP_q$, the $\mathcal{R}(i, S_P)$ are always less than or equal to $\mathcal{R}(i, S_\omega)$ because of $S_P \subset S_\omega$ (*i.e.*, $\frac{\mathcal{R}(i,S_P)}{\mathcal{R}(i,S_\omega)} \leqslant 1$).

For ASP, we hope it can evaluate the errors between semantic ranking $\mathcal{R}(i, C)$ and similarity ranking $\mathcal{R}(i, S)$ to compute the ranking precision. There are two methods to compute the errors, the first is $|\mathcal{R}(i,C) - \mathcal{R}(i,S)|$, and the second is $\frac{\mathcal{R}(i,C)}{\mathcal{R}(i,S)}$. However, different from $AP$ ($\frac{\mathcal{R}(i,S_P)}{\mathcal{R}(i,S_\omega)} \leqslant 1$), $\frac{1}{|S|} \leqslant \frac{\mathcal{R}(i,C)}{\mathcal{R}(i,S)} \leqslant |S|$ makes it hard to calculate the errors with the same scale. Thus, we employ the form $\frac{\mathcal{M}in(\mathcal{R}(i,C), \mathcal{R}(i,S))}{\mathcal{M}ax(\mathcal{R}(i,C), \mathcal{R}(i,S))}$ to convert the definition domain from $(\frac{1}{|S|}, |S|)$ to $(0, 1)$.

---

*Corresponding author.

36th Conference on Neural Information Processing Systems (NeurIPS 2022).

Additionally, compared with the first method ($|\mathcal{R}(i,C) - \mathcal{R}(i,S)|$), the second one ($\frac{Min(\mathcal{R}(i,C),\mathcal{R}(i,S))}{Max(\mathcal{R}(i,C),\mathcal{R}(i,S))}$) is a self-weight method. For example, there are a semantic ranking list $C = \{\mathbf{1}, 3, \mathbf{2}, 5, 4\}$ and a similarity ranking list $S = \{\mathbf{3}, 1, \mathbf{4}, 5, 2\}$. For the first item and the third item (**bold number**), the ranking distance computed by the first method are $|1 - 3| = 2$ and $|2 - 4| = 2$, resulting in the same precision score. Nonetheless, the precision scores computed by the second method are $\frac{Min(1,3)}{Max(1,3)} = \frac{1}{3} < \frac{1}{2} = \frac{Min(2,4)}{Max(2,4)}$, which means that items with high semantic ranking but low similarity ranking will receive more punishment than those with high semantic ranking, even they have the same ranking distance.

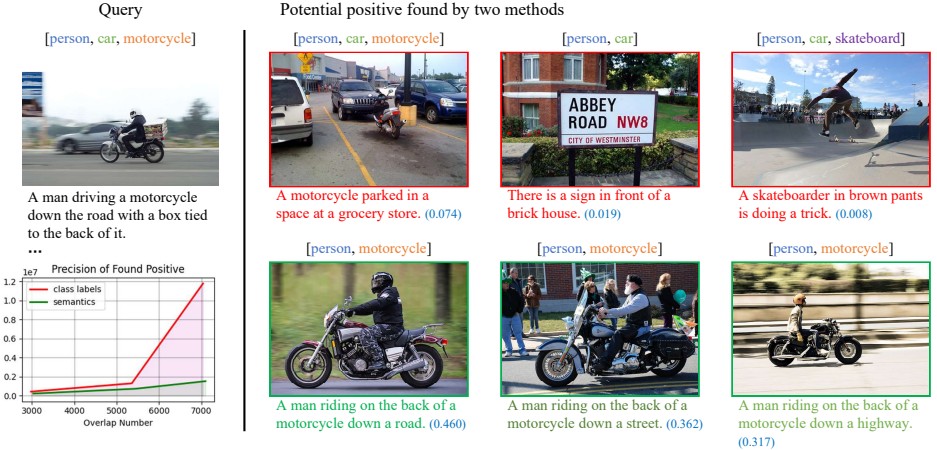

Figure 1: **Comparison of two methods in finding potential positive correspondences.** Given an image as a query, the potential positive correspondences are mined by two methods, class-based and semantics-based. The vector above each image is the class label of the image. The caption below the image comes from the ground truth, followed by the semantic score (computed by CIDEr) between the caption and the ground truth of the query. The positive images found based on class are marked with red boxes, while those found based on semantics are marked with green boxes. The Precision of found positive curves is shown on the left.

.

## B    Comparision with Other Multiplicity Metric

PMRP [1] is proposed to evaluate the multiplicity of the overall retrieval set (including non-ground-truth items) by finding potential positive correspondences based on the class difference. Nonetheless, the class-based method used to find potential positive correspondences brings lots of noise because of low precision (the first row in Figure 1). We employ the CxC dataset [6], which is an extended MS-COCO dataset [4] with more correspondences annotated manually (the number of extended correspondences is about 10000) to validate this shortcoming. Concretely, we count the number of overlapping positive correspondences found by the class-based method in CxC. The curves in Figure 1 show that there are only 7039 samples in CxC while the number of potential correspondences found is about 11.8 million, which shows that PMRP has low accuracy in measuring multiplicity.

Compared with PMRP, ASP is more effective in measuring multiplicity, because semantics-based methods are more accurate in finding potential positive correspondences than class-based methods (the second row in Figure 1). To verify the advantages of ASP over PMRP, we also count the number of overlapping positive correspondences found by the semantics-based method in CxC and report the results in Figure 1.

We can observe that when the number of overlapping positive correspondences is about 7000, the total number of positive correspondences found by the class difference strategy is about 11.8 million, which is 7.7 times as many as found by the semantics (1.5 million). This means that using semantics to find potential correspondences is much more accurate than using class differences.

Although ASP is more accurate than PMRP in measuring multiplicity, there is an inevitable shortcoming. Figure 1 (b) in the main paper shows that the CIDEr [7] scores of a few ground truth captions (about 15%) are less than 0.3. It may bring some noise to the training and evaluation process.

## C The Effect of Different Semantic Metric

Table 1: Ablation studies of different semantic metric on MS-COCO 1K

| | Image-to-text | | | | Text-to-image | | | |
|---|---|---|---|---|---|---|---|---|
| | R@1 | ASP (C) | ASP (B) | ASP (R) | ASP (A) | R@1 | ASP (C) | ASP (B) | ASP (R) | ASP (A) |
| SGR (B) | 77.2 | 53.9 | 53.0 | 52.6 | 53.3 | 61.9 | 54.6 | 53.5 | 52.6 | 53.3 |
| B+CIDEr | **78.0** | **68.5** | 55.7 | 63.3 | 65.1 | **62.6** | **62.8** | 57.9 | 59.1 | **60.8** |
| B+BLEU | 77.6 | 55.6 | **68.7** | 57.8 | 58.7 | 62.0 | 59.2 | **58.9** | 58.2 | 59.6 |
| B+Rouge | 77.4 | 63.2 | 56.6 | **67.1** | **66.0** | 61.9 | 60.5 | 57.9 | **59.4** | 60.5 |

CIDEr is one of the most important metrics to evaluate text similarity in image captioning. Thus, we mainly utilize CIDEr as the semantic metric. To validate the effect of different semantic metrics on DAA, we conduct more experiments using CIDEr [7], BLEU [5], and Rouge [3] on the SGR [2] model. The results are shown in Table 1. Concretely, ASP (C), ASP (B), and ASP (R) are ASP using CIDEr, BLEU, and Rouge as semantic similarity metrics separately. Meanwhile, we utilize the mean of these three difference matrices to compute ASP (A).

As can be seen, SGR retrained with DAA using CIDEr, BLEU, or Rouge as semantic similarity metrics all can obtain performance improvements on R@1 and ASP. Among them, DAA using CIDEr obtains the most significant improvements on R@1, with the highest ASP (A) score (60.8%) in text retrieval and the second highest ASP (A) score (65.1%) in image retrieval. Therefore, compared with BLEU and Rouge, it seems that CIDEr can excavate more precise semantic correspondences.

## D Trained in Different Data Scale

As the number of potential positive relationships increases with the growth of the data scale, it seems that the effectiveness of DAA is related to the data scale. To explore the applicability of DAA in different data scales, we carried out experiments on MS-COCO with 20%, 40%, 60%, 80%, and 100% training data (*i.e.*, using 1, 2, 3, 4, and 5 captions per image). We employ DAA to SGR model, which achieves state-of-the-art performance in the non-probabilistic-based model.

The results in Table 2 show that the SGR trained with DAA invariably emerges with higher performance than the original SGR in different data ratios. In particular, the SGR model does not converge when the data ratio is 20%, but it obtained high performance by training with DAA. Additionally, the lower the data ratio, the larger the performance gap (*e.g.*, 3.5% for 40% data ratio and 0.8% for 100% data ratio on i2t on MS-COCO) between them on R@1, as well as the smaller gap (but still significant, 12.9% for 40% data ratio on i2t, 14.6% for 100% data ratio on i2t) between them on ASP. This proves that the potential positive correspondences could absolutely boost the models' performance.

Additionally, we found that the lower the data ratio, the greater the performance gap between them on R@1, as well as the smaller gap between them on ASP. For this phenomenon, we give an analysis below:

(i) As the models trained with rare data always have poor performance (*i.e.*, the first retrieved items are easy to have a low semantic correlation with queries), the effect of DAA, which could be regarded as a data augmentation approach, is great for these models with poor performance. Therefore, DAA could significantly enhance the performance of R@1 in datasets with insufficient data.

(ii) However, models trained on larger datasets have strong visual and language understanding abilities (*i.e.*, in most retrieval lists, the first retrieved items always have high semantic correlations with the queries, even if it is not in ground truth), more potential positive correspondences mined (data augmentation) in larger datasets are hard to boost R@1 significantly like in small datasets. They pay more attention to improving the multiplicity of the model. Thus, the higher the data ratio is, the higher the ASP score is.

Table 2: Ablation studies of different training data ratio

| Data Ratio | Method | 1K Test Images | | | | 5K Test Images | | | |
|---|---|---|---|---|---|---|---|---|---|
| | | i2t | | t2i | | i2t | | t2i | |
| | | R@1 | ASP | R@1 | ASP | R@1 | ASP | R@1 | ASP |
| 20% | SGR | - | - | - | - | - | - | - | - |
| | SGR+DAA | **68.0** | **61.8** | **53.2** | **60.0** | **44.0** | **61.7** | **32.0** | **59.8** |
| 40% | SGR | 69.2 | 54.1 | 54.7 | 55.1 | 45.4 | 54.0 | 33.1 | 54.9 |
| | SGR+DAA | **72.7** | **67.0** | **58.0** | **61.8** | **50.5** | **66.8** | **36.5** | **61.6** |
| 60% | SGR | 73.0 | 54.5 | 57.7 | 55.0 | 50.5 | 54.3 | 35.6 | 54.8 |
| | SGR+DAA | **74.8** | **68.0** | **60.5** | **62.3** | **52.6** | **67.9** | **38.9** | **62.1** |
| 80% | SGR | 75.0 | 54.0 | 59.5 | 55.1 | 52.3 | 53.9 | 37.7 | 54.8 |
| | SGR+DAA | **76.4** | **68.4** | **61.8** | **62.6** | **54.7** | **68.3** | **40.2** | **62.4** |
| 100% | SGR | 77.2 | 53.9 | 61.9 | 54.6 | 55.8 | 53.7 | 40.1 | 54.4 |
| | SGR+DAA | **78.0** | **68.5** | **62.6** | **62.8** | **56.5** | **68.4** | **40.8** | **62.6** |

In conclusion, the proposed method can improve both R@1 and ASP in multiple scale datasets. In datasets without sufficient data, the performance of R@1 and ASP could be enhanced significantly by using DAA. Nonetheless, in the datasets with a huge amount of data, the gap of R@1 between the model trained with or without DAA is small, but the model trained using DAA will get an impressive performance of ASP and multiplicity.