# OpenReview forum: "A Differentiable Semantic Metric Approximation in Probabilistic Embedding for Cross-Modal Retrieval"
_NeurIPS.cc/2022/Conference — NeurIPS 2022 Accept_

### Official Review · Reviewer_YuA1 · 2022-07-05

**Rating:** 5
**Confidence:** 5
**Soundness:** 2 fair
**Presentation:** 2 fair
**Contribution:** 2 fair

**Summary:**

This paper focuses on addressing a critical problem in probabilistic embedding for cross-modal retrieval, i.e., the multiplicity performance of the model is not well studied due to the not-well-aligned datasets and the current widely-used metrics. To address this problem, the metric Average Semantic Precision (ASP) is defined to evaluate the performance of multiplicity in these not-well-aligned datasets. Moreover, a Differentiable ASP Approximation (DAA) method is proposed to make the model diversified during model training over these noisy datasets. Experiments are conducted to show the effectiveness of our DAA for boosting the multiplicity of probabilistic embedding for cross-modal retrieval.

**Questions:**

See my questions in weaknesses.

**Ethics Review Area:**

["I don’t know"]

**Limitations:**

The limitations of this paper are not discussed.

**Strengths And Weaknesses:**

Strengths:
+ Average Semantic Precision (ASP) is defined to evaluate the performance of multiplicity.
+ A Differentiable ASP Approximation (DAA) method is proposed for model training.

Weaknesses:
- The comparison in Table 1 is far from sufficient. Other baselines should be included to obtain more solid results. See some examples below:\
[a] Learning fragment self-attention embeddings for image-text matching, ACM-MM 2019.\
[b] IMRAM: Iterative Matching With Recurrent Attention Memory for for cross-modal image-text retrieval, CVPR 2020.\
[c] Similarity reasoning and filtration for image-text matching, AAAI 2021.
- The study on multiplicity in this paper is rather shallow. What is the multiplicity performance of an image-text pre-training model for cross-modal retrieval? When a cross-modal model is pre-trained with a very large dataset, can its multiplicity performance still be boosted by the proposed method?
- The convergence property of min L_{ASP} is not well studied. Considering the complexity of this loss function, is the training process via the proposed method stable? Multiple trials with different random seeds should be provided.

---

> ### Author Response · Authors · 2022-08-02
> **Response to Reviewer YuA1**
>
> We would like to thank the reviewer for providing valuable feedback and raising constructive suggestions. In what follows, we wish to address some of the questions you raised as below.
>
> ***Q1: Other baselines should be included to obtain more solid results.***
>
> **A1:**  We compare our approaches with SGRAF [1], which is the best performance model among the three references given. Additionally, SGRAF achieves state-of-the-art performance in non-probabilistic approaches. The results are shown in the **Tab.4** in **Common Response**. It can be seen that our approach still obtains consistent improvements on R@1 and ASP, which means that our approach can also effectively boost the performance of non-probabilistic models with high-performance.
>
> ***Ref:***
>
> [1] Diao, et al. "Similarity Reasoning and Filtration for Image-Text Matching." In AAAI 2021.
>
> ---
>
> ***Q2: Can the multiplicity performance of the model pre-trained with a very large dataset still be boosted by the proposed method?***
>
> **A2:**  Limited by time, it is hard to train the model in a larger dataset than MS-COCO, but we conduct the experiments on MS-COCO with 20%, 40%, 60%, 80%, and 100% training data (_i.e._, using 1, 2, 3, 4 and 5 caption per image) to explore the effect of DAA in different data scales. The results reported in **Tab.6** in **Common Response**.
>
> It can be seen that the SGR trained with DAA invariably emerges with higher performance than the original SGR in different data ratios. Additionally, we found that the lower the data ratio, the greater the performance gap between them on R@1, as well as the smaller gap between them on ASP. For this phenomenon, we give an analysis as below:
>
> - As the models trained with rare data always have poor performance (_i.e._, the first retrieved items are easy to have a low semantic correlation with queries), the effect of DAA, which could be regarded as a data augmentation approach, is great for these models with poor performance. Therefore, DAA could significantly enhance the performance of R@1 in datasets with insufficient data.
>
> - However, models trained on larger datasets have strong visual and language understanding abilities (_i.e._, in most retrieval lists, the first retrieved items always have high semantic correlations with the queries, even if it is not in ground truth), more potential positive correspondences mined (data augmentation) in larger datasets are hard to boost R@1 significantly like in small datasets. They pay more attention to improving the multiplicity of the model. Thus, the higher the data ratio is, the higher ASP score is.
>
> In conclusion, the proposed method can improve both R@1 and ASP in multiple scale datasets. In datasets without sufficient data, the performance of R@1 and ASP could be enhanced significantly by using DAA. Nonetheless, in the datasets with huge amount of data, the gap of R@1 between the model trained with or without DAA is little, but the model trained using DAA will get an impressive performance of ASP and multiplicity. This fully proves that DAA can boost the multiplicity of the model significantly, especially in datasets with large amounts of data.
>
> ---
>
> ***Q3: Multiple trials with different random seeds should be provided to validate the convergence property.***
>
> **A3:**  To prove the convergence property of $\mathcal Min(\mathcal L_{ASP})$, as well as the reproducibility of reported results, we repeat the experiments 5 times with different random seeds and plot the curve of R@1 against the epoch in **"README.md"** of the link (https://anonymous.4open.science/r/2022-NeurIPS-DAA-4F1F). The best results of the five experiments are shown in the table below. It proves that the object function proposed has good convergence property, and the results reported can be reproduced stably.
>
> |     |      |     | 1K   | Images |     |      |   |   |      |     | 5K   | Images |     |      |
> |-----|------|-----|------|--------|-----|------|---|---|------|-----|------|--------|-----|------|
> |     |      | **i2t** |      |        | **t2i** |      |   |   |      | **i2t** |      |        | **t2i** |      |
> | **No.** | **R@1**  |     | **ASP**  | **R@1**    |     | **ASP**  |   |   | **R@1**  |     | **ASP**  | **R@1**    |     | **ASP**  |
> | 1   | 71.2 |     | 58.4 | 57.8   |     | 57.2 |   |   | 48.2 |     | 58.3 | 35.1   |     | 57.0 |
> | 2   | 71.1 |     | 58.4 | 57.7   |     | 57.2 |   |   | 47.9 |     | 58.3 | 35.2   |     | 57.0 |
> | 3   | 70.6 |     | 58.4 | 57.8   |     | 57.2 |   |   | 47.7 |     | 58.3 | 34.8   |     | 57.0 |
> | 4   | 70.3 |     | 58.4 | 57.6   |     | 57.2 |   |   | 47.8 |     | 58.3 | 34.9   |     | 57.0 |
> | 5   | 70.3 |     | 58.4 | 57.8   |     | 57.2 |   |   | 47.2 |     | 58.2 | 34.8   |     | 57.0 |

---

> > ### Comment · Reviewer_YuA1 · 2022-08-10
> > **Response to authors' rebuttal**
> >
> > Thanks for the detailed rebuttal. My concerns have been addressed. I have raised my rating.

---

> ### Author Response · Authors · 2022-08-07
> **We are happy to address any further concerns**
>
> Dear Reviewer YuA1,
>
> We sincerely appreciate the reviewer's effort and constructive comments. We have carried out more experiments and provided detailed responses to clarify all questions. We hope you are satisfied with our answers. If you have any further concerns, please feel free to let us know and we are more than happy to address them.
>
> Best wishes!
>
> Yours,
>
> Authors of Paper 5175

---

### Official Review · Reviewer_vghc · 2022-07-10

**Rating:** 5
**Confidence:** 5
**Soundness:** 3 good
**Presentation:** 2 fair
**Contribution:** 2 fair

**Summary:**

This paper focuses on the many-to-many relationships in the image-text retrieval task. The authors notice that text pairs with high CIDEr scores are more likely relevant, and thus the corresponding image-text pairs should be considered as positive. On top of this, they propose a metric named Average Semantic Precision (ASP) to capture this information. Then a differentiable surrogate loss of ASP is proposed to directly optimize ASP.

**Questions:**

Please refer to the weaknesses part for my main concerns. Besides, I still have some minor concerns:
1. The example in Fig.1(a) shows that the ground truth captions might have lower CIDEr scores. Is it reasonable to punish the model when a ground truth caption ranked higher than captions with higher CIDEr scores?
2. According to the motivation of this paper, some proper captions might be missed in the annotations. This phenomenon also occurs in the test data, i.e., these missing captions are not considered in the metrics like PMRP and R@1. Could you please explain why the proposed method can still achieve performance improvements on these inconsistent metrics?


**Limitations:**

Yes

**Strengths And Weaknesses:**

The main strength of this paper is the observation that CIDEr scores are useful to measure the many-to-many relationships. It could be helpful for image-text retrieval, especially in the self-supervised setting.

However, the paper could be further improved in the following aspects:
1. Although the proposed ASP loss looks reasonable to align the prediction with the CIDEr scores, the motivation of the specific formulation (Eq.(4)) should be clarified.
2. The probabilistic embedding part is exactly the same as the work of Chun et al. It seems that Sec.3 has nothing to do with the proposed ASP. Moreover, it is unclear how these two components are combined in this paper.
3. Some notations are confusing:
- In Eq.(8), the $U$ in $R(i, U)$ is an index set, but both $S$ and $C$ in Eq.(4) are score sets;
- In Eq.(11), the sigmoid function is applied to a Boolean variable: $\phi(D_{ij}^C > 0)$, should it be $\phi(-D_{ij}^C)$? Also, should $I{D_{ij}^C > 0}$ be $I{D_{ij}^C < 0}$?
4. The provided code repository is empty.
5. All results of competitors are directly copied from other papers, but some implementation details in this paper are different from the original papers of competitors: the training process of PCME consists of two phases including a warmup phase (only train the head modules) for 30 epochs and a finetune phase (train all parameters) for 30 epochs. While the proposed method is trained for three phases including a warmup phase for 3 epochs, a pre-training phase (train the head module of the visual encoder and all parameters of the extual encoder) for 40 epochs, and a finetune phase for 27 epochs. It looks unfair to compare these results under different training settings. Moreover, the result of VSE++ in MS-COCO 5K T2I should be 30.3 instead of 31.3.
6. The details of baseline in ablation studies are unclear. The baseline in Tab.3 outperforms competitors in Tab.1, could you please explain the reasons for the performance gain?

=============================================

After rebuttal, most of my concerns have been addressed, and I'd like to raise my rating to 5. However, I still feel that the introduction of the probabilistic embedding is unnecessary since the proposed method can be directly applied to other cross-modal methods. On the contrary, the motivation of Eq.(4) is more important and interesting. Therefore, it might be better to state the motivation in the main paper instead of the probabilistic embedding.

---

> ### Author Response · Authors · 2022-08-02
> **[2/2] Response to Reviewer vghc**
>
> ***Q5: The provided code repository is empty.***
>
> **A5:** We're sorry for taking so long to publish the code. Now, we have released the code of PCME using DAA and the code of SGRAF using DAA. Additionally, we also publish the evaluation files with the results reported in the paper at the link in the abstract (https://anonymous.4open.science/r/2022-NeurIPS-DAA-4F1F).
>
> To prove the reproducibility of reported results. We repeat the experiments 5 times with different random seeds and plot the curve of R@1 against the epoch in **"README.md"** of the link. The best results of the five experiments are also shown in the link.
>
> ---
>
> ***Q6: It is unfair to compare the results under different training settings.***
>
> **A6:** We retrain PCME (using bert as textual encoder) with the same training strategy as of proposed method and report the results as PCME-bert in **Tab.1** and **Tab.2** in the **paper**.
>
> To validate the effectiveness of DAA, we apply the proposed method to more baselines, such as SGRAF, which achieves state-of-the-art performance in non-probabilistic models. The results are reported in **Tab.4** in **Common Response**. All experiments keep the same setting. As can be seen, DAA is effective to improve the performance of R@1 and ASP in multiple baselines.
>
> ---
>
> ***Q7: The result of VSE++ in MS-COCO 5K T2I should be 30.3 instead of 31.3.***
>
> **A7:** Thanks for your detailed review. We have modified it in the updated paper.
>
> ---
>
> ***Q8: Why the baseline in Tab.3 outperforms competitors in Tab.1***
>
> **A8:** In **Tab.1** in the **paper**, to fairly show the advantages of using Bert as a textual encoder over using GRU as a textual encoder in probabilistic embedding, we employ the number of samples in PCME-bert consistent with that in PCME (N=7). However, we find that PCME-bert achieves the best performance when N=5, so the number of samples used in our proposed method is 5. Meanwhile, to truthfully and quantitatively analyze the performance improvement brought by the DAA module, we show the results of baseline with N=5 in **Tab.3** in the **paper**. We are sorry to make confusion to you, we have added explanations and modified results in the revised paper.
>
> ---
>
> ***Q9: Is it reasonable to punish the model when a ground truth caption ranked higher than captions with higher CIDEr scores?***
>
> **A9:** The CIDEr score is computed based on the five ground truth captions. Thus, a ground truth caption is easy to obtain a high CIDEr score because it will compute the CIDEr score with itself. If a ground truth caption has a lower CIDEr score than other non-ground-truth captions, it means that this ground truth caption deviates from the other four ground truth captions, and the non-ground-truth caption is more similar to the whole ground truth in semantics than deviated ground-truth caption.
>
> Additionally, we count the distribution of CIDEr scores of ground truth captions in MS-COCO 5K sets (which contain 25000 positive image-text pairs) and find that the CIDEr scores of 95.5% positive image-text pairs are more than 0.25. Inevitably, there are some noises in these potential positive correspondences found by the proposed method, but the positive impact of finding these potential correspondences on the performance of the model is far greater than the negative impact of these noises. The penalty has little impact on the performance of the model.
>
> ---
>
> ***Q10: Why the proposed method can still achieve performance improvements on inconsistent metrics (_e.g._, PMRP, R@1)?***
>
> **A10:** There are two reasons as follows:
> - DAA can find potential positive correspondences, which could be regarded as an approach to data augmentation. It can improve the performance of the model on PMRP and R@1.
>
> - The ground-truth captions always have a high CIDEr score with a high ranking. Furthermore, in almost all batches, ground truth captions rank 1 in semantics while training because of shuffle sampling. These make the ranking of ground truth can be directly optimized by the proposed ranking-based method. Thus, the ranking-based metric PMRP and R@1 can consistently obtain improvements.

---

> ### Author Response · Authors · 2022-08-02
> **[1/2] Response to Reviewer vghc**
>
> We thank the reviewer for the patient reading and constructive feedback towards improving our manuscript. In what follows, we wish to address some of the questions you raised as below.
>
> ***Q1: The motivation of the specific formulation (Eq.(4)) should be clarified.***
>
> **A1:**  We design the formulation of ASP (Eq.(4)) with reference to AP (Average Precision).
>
> $$AP_q=\frac{1}{\left| S_P\right|}\sum_{i \in \omega}\frac{\mathcal R(i,S_P)}{\mathcal R(i,S_\omega)}$$
>
> where $S_P=\\{s_\zeta, \zeta \in \mathcal P_q\\}, S_N=\\{s_\xi, \xi \in \mathcal N_q\\}$ are the positive and negative relevance score sets, respectively. Addtionally, $S_\omega = S_P \cup S_N$.
>
> Intuitively, $\frac{\mathcal R(i,S_P)}{\mathcal R(i,S_\omega)}$ is the distance between the ranking of $i$-th item in $S_P$ and the ranking of it in $S_\omega$. Meanwhile, in $AP_q$, the $\mathcal R(i,S_P)$ are always smaller than $\mathcal R(i,S_\omega)$ because of $S_P \subset S_\omega$ (_i.e._, $\frac{\mathcal R(i,S_P)}{\mathcal R(i,S_\omega)} < 1$).
>
> For ASP, we hope it can evaluate the errors between semantic ranking $\mathcal R(i, C)$ and similarity ranking $\mathcal R(i, S)$ to compute the ranking precision. There are two methods to compute the errors, the first is $\left |  \mathcal R(i,C) - \mathcal R(i,S)\right |$, and the second is $\frac{\mathcal R(i,C)}{\mathcal R(i,S)}$. However, different from AP ($\frac{\mathcal R(i,S_P)}{\mathcal R(i,S_\omega)} < 1$), $\frac{1}{i}< \frac{\mathcal R(i,C)}{\mathcal R(i,S)} < i$ makes it hard to calculate the errors with the same scale. Thus, we employ the form $\frac{\mathcal Min(\mathcal R(i,C),\mathcal R(i,S))}{\mathcal Max(\mathcal R(i,C),\mathcal R(i,S))}$ to convert the definition domain from $(\frac{1}{i}, i)$ to $(0, 1)$.
>
> Additionally, compared with the first method ($\left | \mathcal R(i,C) - \mathcal R(i,S)\right |$), the second one ($\frac{\mathcal Min(\mathcal R(i,C),\mathcal R(i,S))}{\mathcal Max(\mathcal R(i,C),\mathcal R(i,S))}$) is a self-weight method. For example, there are a semantic ranking list $C=\\{\pmb{1}, 3, \pmb{2}, 5, 4\\}$ and a similarity ranking list $S=\\{\pmb{3}, 1, \pmb{4}, 5, 2\\}$. For the first item and the third item (**bold number**), the ranking distance computed by the first method are $|1-3|=2$ and $|2-4|=2$, resulting in the same precision score. Nonetheless, the precision scores computed by the second method are $\frac{\mathcal Min(1,3)}{\mathcal Max(1,3)}=\frac{1}{3} < \frac{1}{2} = \frac{\mathcal Min(2,4)}{\mathcal Max(2,4)}$, which means that items with high semantic ranking but low similarity ranking will receive more punishment than those with high semantic ranking, even they have the same ranking distance.
>
> ---
>
> ***Q2: It is unclear how the probabilistic embedding part is combined with the proposed method in this paper.***
>
> **A2:** Diversity is a serious problem in probabilistic embedding, so we give a brief introduction in Sec.3 to probabilistic embedding. In fact, our proposed method is not only suitable for probabilistic embedding but can also be integrated into any two-stream cross-modal retrieval model.
>
> Compared with non-probabilistic embedding, the only difference in applying ASP to probabilistic embedding is the multiple samples. Concretely, given 64 image-text pairs in a batch with sample number 5, the probabilistic model will obtain a similarity score matrix with shape (320, 320). The shape of the semantic matrix is also (320, 320), all samples from the same image-text pairs will obtain the same semantic scores. Then, ASP will be computed based on these two matrices by Eq.(11).
>
> ---
>
> ***Q3: In Eq.(8), the $U$ in $R(i, U)$ is an index set, but both $S$ and $C$ in Eq.(4) are score sets.***
>
> **A3:**
> In Eq.(8),
> $$\mathcal R(i,U)=1+\sum_{j \in U,j \neq i}\mathbb{I}\{(u_i-u_j)<0\},$$
> Actually, $U$ is one of $C$ or $S$, _i.e._, the Eq.(8) can be written as:
> $$\mathcal R(i,C)=1+\sum_{j \in C,j \neq i}\mathbb{I}\{(c_i-c_j)<0\},$$
> or
> $$ \mathcal R(i,S)=1+\sum_{j \in S,j \neq i}\mathbb{I}\{(s_i-s_j)<0\}.$$
>
> ---
>
> ***Q4: In Eq.(11), $\phi(D^C_{ij}>0)$ should be $\phi(-D^C_{ij})$. Also, should $\mathbb{I}\\{D^C_{ij}>0\\}$ be $\mathbb{I}\\{D^C_{ij}<0\\}$?***
>
> **A4:** In Eq.(11), $\phi(D^C_{ij}>0)$ is indeed incorrect, we are sorry to make confusion to you, the mistake has been revised in the paper, but it should be $\phi(D^C_{ij})$ instead of $\phi(-D^C_{ij})$, the reason is as follow:
>
> In Eq.(9),
>
>  $$D^U={\begin{bmatrix}
>  u_1 & \cdots & u_m \\\\
>  \vdots& \ddots & \vdots \\\\
>  u_1 & \cdots & u_m \\\\
> \end{bmatrix}}
> -{\begin{bmatrix}
>  u_1 & \cdots & u_1 \\\\
>  \vdots& \ddots & \vdots \\\\
>  u_m & \cdots & u_m \\\\
> \end{bmatrix}}$$
>
> Thus, $D^U_{ij} = u_j -u_i$ (_e.g._, $D^U_{12} = u_2 -u_1$). The Indicator function will select 1 while $u_i -u_j <0$ (Eq.(8)), _i.e._, it will select 1 while $D^U\_{ij} = u_j - u_i > 0$. Therefore, the form in Eq.(10) should be the $\mathbb{I}\\{D^C\_{ij}>0\\}$ and the $\phi(D^C\_{ij}>0)$ should be $\phi(D^C\_{ij})$ in Eq.(11).

---

> ### Author Response · Authors · 2022-08-07
> **We are happy to address any further concerns**
>
> Dear Reviewer vghc,
>
> We sincerely appreciate the reviewer's effort and constructive comments. We have conducted more experiments and provided detailed responses to clarify all questions. If you have any further concerns, please feel free to let us know and we are more than happy to answer them.
>
> Best wishes!
>
> Yours,
>
> Authors of Paper 5175

---

### Official Review · Reviewer_mhZH · 2022-07-12

**Rating:** 8
**Confidence:** 4
**Soundness:** 3 good
**Presentation:** 3 good
**Contribution:** 3 good

**Summary:**

This paper addresses the multiplicity of correspondences between images and texts in cross-modal retrieval. Firstly, they utilize semantic correlation computed by CIDEr to construct the potential correspondence. Then they propose a new metric, named Average Semantic Precision (ASP) to measure the ranking precision of semantic correlation for retrieval sets. Finally, they propose to directly optimize ASP in their objective function directly by making the ranking function of ASP differentiable through a sigmoid function. Experimental results show the promising performance of the proposed method.

**Questions:**

Please see Weaknesses

**Limitations:**

Yes

**Strengths And Weaknesses:**

The work seems to be technically sound; key modeling assumptions are clearly stated, claims are nicely demonstrated and supported by the results.
1) Unlike existing cross-modal retrieval methods, which treat the multiplicity of correspondences between images and texts in cross-modal retrieval task as a multi-label problem, this paper considers it as a ranking problem, which is quite interesting, and more reasonable.
2) They propose a new evaluation metric named Average Semantic Precision (ASP) to measure the ranking precision of semantic correlation for retrieval sets.
3) It is novel to directly optimize the ASP in their designed network.
4) The proposed method outperforms SOTA on ASP, and also outperforms them on R@1 (3.0% for i2t and 1.2% for t2i on MS-COCO).

I believe that this paper will be a valuable contribution to the field, and I recommend acceptance. Below I have a few questions and comments for the authors.
1) Why do you build your method based on 'Probabilistic Embedding'? It seems that the proposed model can be integrated into any cross-modal retrieval method. Therefore, it is necessary to evaluate the proposed model on other cross modal methods instead of 'Probabilistic Embedding'.
2) Similarly, this model utilizes semantic correlation computed by CIDEr to construct the potential correspondence. It is also necessary to evaluate this model based on other methods e.g., Bert instead of CIDEr.
3) Writing is a weak point of this paper. There are lots of typos and grammar errors.
->LINE 284. there are only few improvements on R@1
->LINE 302. Meanwhile, the curves of Smooth-AP and baseline are almost consistent, which shows that only focusing on ground truth is limited to 302 improving multiplicity.
->The texts in Fig.3 and 4 are too small
-> The references format is inconsistent
4) Another major concern is about the proposed ASP metric. Since it is based on ranking, what is the difference between ASP and existing ranking-based evaluation metrics?

---

> ### Author Response · Authors · 2022-08-02
> **Response to Reviewer mhZH**
>
> We thank the reviewer for the positive and detailed review, as well as the suggestions for improvement. Our responses to the reviewer’s questions are below:
>
> ***Q1：It is necessary to evaluate the proposed model on other cross-modal methods instead of 'Probabilistic Embedding'.***
>
> **A1:** Thanks for your contructive suggestions. We integrate our model to SGRAF, which achieves state-of-the-art performance in the model without large-scale pre-training. The results are shown in **Tab.4** in **Common Response**. It can be seen that our approach still obtains consistent improvements on R@1 and ASP, which means that our approach can also effectively boost the performance of non-probabilistic models with high-performance.
>
> ---
>
> ***Q2：It is necessary to evaluate this model based on other semantic metrics instead of CIDEr.***
>
> **A2:**  CIDEr is one of the most commonly used metrics to evaluate text similarity in image captioning. Thus, we mainly utilize CIDEr as the semantic metric in the paper. To validate the effect of different semantic metric to DAA, we extra utilize BLEU and Rouge (two caption similarity metrics used in image captioning like CIDEr) as semantic metrics instead of CIDEr. The results are shown in **Tab.5** in **Common Response**. Concretely, ASP (C), ASP (B), and ASP (R) are ASP using CIDEr, BLEU, and Rouge as semantic similarity metrics, separately. Meanwhile, we utilize the mean of these three difference matrices to compute ASP (A). As can be seen, SGR retrained with DAA using CIDEr, BLEU, or Rouge as semantic similarity metrics all can obtain performance improvements on R@1 and ASP. Among them, DAA using CIDEr obtains the most significant improvements on R@1, with the highest ASP (A) score (60.8%) in text retrieval and the second highest ASP (A) score (65.1%) in image retrieval. It seems that CIDEr can excavate more precise semantic correspondences than others.
>
> ---
>
> ***Q3：There are lots of typos and grammar errors.***
>
> **A3:** Thanks for your detailed review. We have modified these errors in the revised paper.
>
> ---
>
> ***Q4：What is the difference between ASP and existing ranking-based evaluation metrics?***
>
> **A4:** There are some ranking-based evaluation metrics, such as Rank@K and AP. They only focus on the ranking of ground truth in the retrieved set, resulting in limited evaluation for the whole retrieved items. There are some differences between these two metrics, Rank@K only focuses on the ranking of ground truth items with the highest similarity score in the retrieved set, and AP evaluates the quality of retrieval by computing the ranking precision of all ground truth items in the retrieved set.
>
> Different from them, ASP is a metric that not only focuses on the ground truth, it considers the correlation between non-ground-truth items and ground truth. It can evaluate the ranking precision of each item in the retrieved set.

---

> ### Author Response · Authors · 2022-08-08
> **We are happy to address any further concerns**
>
> Dear Reviewer mhZH,
>
> We sincerely appreciate the reviewer's valuable and constructive comments. We have carried out more experiments and provided detailed responses to clarify all questions. We hope you are satisfied with our answers. If you have any further concerns, please feel free to let us know and we are more than happy to address them.
>
> Best wishes!
>
> Yours,
>
> Authors of Paper 5175

---

### Official Review · Reviewer_CS9A · 2022-07-12

**Rating:** 7
**Confidence:** 5
**Soundness:** 3 good
**Presentation:** 3 good
**Contribution:** 3 good

**Summary:**

The multiplicity of correspondences between images and texts in cross-model retrieval greatly increases the difficulty of this task. To tackle this problem, probabilistic embeddings are proposed to quantify many-to-many relationships, but nonexhaustive annotations limit their performance. Based on this observation, they utilize semantic correlation computed by CIDEr to provide a pseudo annotation. Then they propose an effective metric, named Average Semantic Precision (ASP), to measure the ranking precision of semantic correlation for retrieval sets.
Additionally, they introduce a novel and concise objective, coined Differentiable ASP Approximation (DAA) to optimize ASP directly by making the ranking function of ASP differentiable through a sigmoid function. To verify the effectiveness of their approach, extensive experiments are conducted on MS-COCO and CUB Captions.

**Questions:**

From reading the paper, it is very evident that the authors do not come from ML but from a CV background. Some of the concepts are used in a weird name before definition, e.g., 'multiplicity of correspondences between images and texts', 'semantic correlation', 'many-to-many relationships.' in the abstract.

Methods:
It is not convincing that ASP is a valid evaluation metric. More descriptions and experiments are needed.
What is the benefit of these evaluation metrics, and what are the shortcomings?
It is unclear to me why ASP is stuck to CIDEr?

My major concern is about the experiments. The experiment part has three issues:
Few experiments: I think it is not enough to make general claims from having only two experiment datasets. Especially as the datasets are pretty small, i.e., the two datasets are relatively small, with 123,287 images and 11,788 images. Therefore, different and more datasets would enormously benefit the paper's claims.

Lack of quantitative comparisons. The proposed method improves existing SOTAin what cases? and when it fails? Why can it benefit both ASP and R@1?

Missing comparison with alternative approaches. This paper compares with VSE0 [11], PVSE[38], and PCME[7], which were published in 2017, 2019, and 2021.  There is some substantial literature on cross-modal retrieval recently.  Therefore, the paper should experimentally evaluate how these models compare to the proposed approach.

**Limitations:**

See above

**Strengths And Weaknesses:**

This work nicely combines a novel evaluation metric for cross-model retrieval and deep learning model in a novel way and provides an effective approach. The paper is also well-motivated and well-organized.
Major comments:
i) They find that incompletely annotating MPPs is a crucial factor restricting many-to-many mapping for probabilistic embedding. Thus, we propose an effective evaluation metric named ASP, instead of the existing ones, which are powerless to reflect the correlations between each retrieved item and the query.
ii) To eliminate the adverse effect of MPPs in the training process, they propose a novel and concise metric learning method DAA, which approximates and optimizes ASP directly. This seems to be one of the major contributions of this paper.
iii) Promising experimental results.

I believe the paper has some merits. However, it also faces serval weaknesses that need to be addressed.

---

> ### Author Response · Authors · 2022-08-02
> **[2/2] Response to Reviewer CS9A**
>
> ***Q4 : It is not enough to make general claims from having only two experiments datasets.***
>
> **A4:** We employ SGRAF trained with the proposed method on Flicker30K, which is a commonly used dataset in cross-modal retrieval. The results listed below show that our proposed method is also effective to improve the performance of the model trained in multiple datasets.
>
> |           |      | i2t |      |        | t2i |      |
> |-----------|------|-----|------|--------|-----|------|
> | **Method**    | **R@1**  |     | **ASP**  | **R@1**    |     | **ASP**  |
> | SAF*      | 73.8 |     | 49.0 | 56.7   |     | 50.0 |
> | SAF+DAA   | **73.9** |     | **65.0** | **56.9**   |     | **58.4** |
> | SGR*      | 73.7 |     | 54.6 | 55.8   |     | 55.1 |
> | SGR+DAA   | **73.8** |     | **65.5** | **56.6**   |     | **59.0** |
> | SGRAF*    | 77.1 |     | 49.4 | 58.5   |     | 50.6 |
> | SGRAF+DAA | **78.0** |     | **65.8** | **59.9**   |     | **59.2** |
>
> "**\***" denotes the model retrained by public codes.
>
> ---
>
> ***Q5 : Why can proposed method benefit both ASP and R@1?***
>
> **A5:** There are two reasons that DAA also benefits R@1 as follows:
> - DAA is able to find potential positive correspondences, which could be regarded as an approach to data augmentation, so it can improve the performance of PMRP and R@1.
> - The ground-truth captions always have high CIDEr scores with a high ranking. Furthermore, in almost all batches, ground truth captions rank 1 in semantics while training because of shuffle sampling. These make the ranking of ground truth can be directly optimized by the proposed ranking-based method. Thus, the ranking-based metric PMRP and R@1 can consistently obtain improvements.
>
> ---
>
> ***Q6 : The proposed method improves existing SOTA in what cases? and when it fails?***
>
> **A6:** As the number of potential positive relationships increases with the growth of the data scale, it seems that the effectiveness of DAA is related to the data scale. To explore the applicability of DAA in different data scales, we carried out experiments on MS-COCO with 20%, 40%, 60%, 80%, and 100% training data (_i.e_., using 1, 2, 3, 4, and 5 captions per image). To reduce the time cost, we employ DAA to SGR model, which achieves state-of-the-art performance in the model using non-probabilistic embedding. The results are shown in **Tab.6** in **Common Response**.
>
> It can be seen that the SGR trained with DAA invariably emerges with higher performance than the original SGR in different data ratios.  This proves that the potential positive correspondences could absolutely boost the performance of the model.
>
> Additionally, we found that the lower the data ratio, the greater the performance gap between them on R@1, as well as the smaller gap between them on ASP. For this phenomenon, we give an analysis as below:
>
> - As the models trained with rare data always have poor performance (_i.e._, the first retrieved items are easy to have a low semantic correlation with queries), the effect of DAA, which could be regarded as a data augmentation approach, is great for these models with poor performance. Therefore, DAA could significantly enhance the performance of R@1 in datasets with insufficient data.
>
> - However, models trained on larger datasets have strong visual and language understanding abilities (_i.e._, in most retrieval lists, the first retrieved items always have high semantic correlations with the queries, even if it is not in ground truth), more potential positive correspondences mined (data augmentation) in larger datasets are hard to boost R@1 significantly like in small datasets. They pay more attention to improving the multiplicity of the model. Thus, the higher the data ratio is, the higher ASP score is.
>
> In conclusion, the proposed method can improve both R@1 and ASP in multiple scale datasets. In datasets without sufficient data, the performance of R@1 and ASP could be enhanced significantly by using DAA. Nonetheless, in the datasets with huge amount of data, the gap of R@1 between the model trained with or without DAA is little, but the model trained using DAA will get an impressive performance of ASP and multiplicity.
>
> ---
>
> ***Q7 : The paper should experimentally evaluate how recent models compare to the proposed approach.***
>
> **A7:** We compare our approaches with more cross-modal retrieval methods, such as SGRAF with state-of-the-art performance in non-probabilistic approaches. The results are shown in **Tab.4** in **Common Response**. It can be seen that our approach still obtains a consistent improvement on both R@1 and ASP, which means that our approach can also effectively boost the performance of non-probabilistic models with high-performance.

---

> ### Author Response · Authors · 2022-08-02
> **[1/2] Response to Reviewer CS9A**
>
> We would like to thank the reviewer CS9A for providing valuable feedback and encouraging comments. In what follows, we wish to address some of the questions you raised as below.
>
> ***Q1 : Some of the concepts are used in a weird name before definition in the abstract.***
>
> **A1:** Thanks for the constructive suggestions. We have modified these concepts in the revised paper.
>
> ---
>
> ***Q2 : What is the benefit of these evaluation metrics, and what are the shortcomings?***
>
> **A2:** There are three evaluation metrics mainly used in the paper, R@K, PMRP, and ASP.
> - R@K is a ranking-based metric only focusing on the ranking of ground truth items with the highest similarity score in the retrieved set. It can easily measure the ranking precision of ground-truth. However, only focusing on ground truth makes it unable to evaluate the quality of other non-ground-truth ranking.
> - PMRP is proposed to evaluate the multiplicity of the overall retrieval set (including non-ground-truth items) by finding potential positive correspondences based on the class difference. Nonetheless, the class-based method used to find potential positive correspondences brings lots of noise because of low precision. We employ the CxC dataset [1], which is an extended MS-COCO dataset with more correspondences annotated manually (the number of extended correspondences is about 10000) to validate this shortcoming. Concretely, we count the number of overlapping positive correspondences found by the class-based method in CxC. The results are shown in the **Table** below. It can be seen that there are only 7039 samples in CxC while the number of potential correspondences found is about 11.8 million, which shows that PMRP has low accuracy in measuring multiplicity.
> - Compared with PMRP, ASP is more effective in measuring multiplicity, because semantics-based methods are more accurate in finding potential positive correspondences than class-based methods. To verify the advantages of ASP over PMRP, we also count the number of overlapping positive correspondences found by semantics-based method in CxC. The results are listed below.
>
> |                   |         | semantics |         |   |         | class |          |
> |:-----------------:|:-------:|:---------:|:-------:|:---:|:-------:|:-----:|:--------:|
> | $\text{overlap} \approx$ | **overlap** |           | **sum**     |   | **overlap** |       | **sum**      |
> | 3000              | 3032    |           | 234568  |   | 2976    |       | 440390   |
> | 5400              | 5448    |           | 741630  |   | 5357    |       | 1309520  |
> | 7000              | 7098    |           | 1534059 |   | 7039    |       | 11792860 |
>
> It can be seen that when the number of overlapping positive correspondences is about 7000, the total number of positive correspondences found by the class difference strategy is about 11.8 million, which is 7.7 times as many as found by the semantics (1.5 million). This means that using semantics to find potential correspondences is much more accurate than using class differences.
>
> Although ASP is more accurate than PMRP in measuring multiplicity, there is an inevitable shortcoming. Fig.1 (b) in the paper shows that the CIDEr scores of a few ground truth captions (about 15%) are less than 0.3. It may bring some noise to the training and evaluation process.
>
> ***Ref:***
>
> [1] Parekh, et al. "Crisscrossed Captions: Extended Intramodal and Intermodal Semantic Similarity Judgments for MS-COCO." In EACL 2021.
>
> ---
>
> ***Q3 : Why is ASP only stuck to CIDEr?***
>
> **A3:** CIDEr is one of the most commonly used metrics to evaluate text similarity in image captioning. Thus, we mainly utilize CIDEr as the semantic metric. To validate the effect of different semantic metric to DAA, we conduct more experiments using different semantic metrics (CIDEr, BLEU, and Rouge), and the results are shown in **Tab.5** in **Common Response**. As can be seen, SGR retrained with DAA using CIDEr, BLEU, or Rouge as semantic similarity metrics all can obtain performance improvements on R@1 and ASP. Among them, DAA using CIDEr obtains the most significant improvements on R@1, with the highest ASP (A) score (60.8%) in text retrieval and the second highest ASP (A) score (65.1%) in image retrieval. Therefore, compared with BLEU and Rouge, it seems that CIDEr can excavate more precise semantic correspondences.
>
> ---

---

> ### Author Response · Authors · 2022-08-08
> **We are happy to address any further concerns**
>
> Dear Reviewer CS9A,
>
> We sincerely appreciate the reviewer's valuable and constructive comments. We have carried out more experiments and provided detailed responses to clarify all questions. We hope you are satisfied with our answers. If you have any further concerns, please feel free to let us know and we are more than happy to address them.
>
> Best wishes!
>
> Yours,
>
> Authors of Paper 5175

---

### Author Response · Authors · 2022-08-02
**[2/2] Common Response**

## 3. Trained with Different Data Scale ##

DAA can excavate potential positive correspondences, which can be regarded as a data augmentation approach. Therefore, we explore whether it can help the model be trained well in a dataset with little data. We carried out experiments on MS-COCO with 20%, 40%, 60%, 80%, 100% training data (_i.e._, using 1, 2, 3, 4 and 5 captions per image). To reduce the huge time cost caused by probabilistic embedding, we employ DAA to the SGR model and report the results in **Tab.6** as below.

The results in **Tab.6** shows that the SGR trained with DAA invariably emerges with higher performance than the original SGR in different data ratios. In particular, the SGR model does not converge when the data ratio is 20%, but it obtained high performance by training with DAA. Additionally, the lower the data ratio, the larger the performance gap (_e.g._, 3.5% for 40% data ratio and 0.8% for 100% data ratio on i2t on MS-COCO) between them on R@1, as well as the smaller gap (but still large, 12.9% for 40% data ratio on i2t, 14.6% for 100% data ratio on i2t) between them on ASP.

### ***Table 6*** ###
|            |         |      |     | 1K   |      | Test   |     |      |      |      |      |     | 5K   |      | Test   |     |      |
|------------|---------|------|-----|------|------|--------|-----|------|------|------|------|-----|------|------|--------|-----|------|
|            |         |      | **i2t** |      |      |        | **t2i** |      |      |      |      | **i2t** |      |      |        | **t2i** |      |
| **Data Ratio** | **Method**  | **R@1**  |     | **ASP**  |      | **R@1**    |     | **ASP**  |      |      | **R@1**  |     | **ASP**  |      | **R@1**    |     | **ASP**  |
| 20%        | SGR     | -    |     | -    |      | -      |     | -    |      |      | -    |     | -    |      | -      |     | -    |
|            | SGR+DAA | **68.0** |     | **61.8** |      | **53.2**   |     | **60.0** |      |      | **44.0** |     | **61.7** |      | 32.0   |     | 59.8 |
| 40%        | SGR     | 69.2 |     | 54.1 |      | 54.7   |     | 55.1 |      |      | 45.4 |     | 54.0 |      | 33.1   |     | 54.9 |
|            | SGR+DAA | **72.7** |     | **67.0** |      | **58.0**   |     | **61.8** |      |      | **50.5** |     | **66.8** |      | **36.5**   |     | **61.6** |
| 60%        | SGR     | 73.0 |     | 54.5 |      | 57.7   |     | 55.0 |      |      | 50.5 |     | 54.3 |      | 35.6   |     | 54.8 |
|            | SGR+DAA | **74.8** |     | **68.0** |      | **60.5**   |     | **62.3** |      |      | **52.6** |     | **67.9** |      | **38.9**   |     | **62.1** |
| 80%        | SGR     | 75.0 |     | 54.0 |      | 59.5   |     | 55.1 |      |      | 52.3 |     | 53.9 |      | 37.7   |     | 54.8 |
|            | SGR+DAA | **76.4** |     | **68.4** |      | **61.8**   |     | **62.6** |      |      | **54.7** |     | **68.3** |      | **40.2**   |     | **62.4** |
| 100%       | SGR     | 77.2 |     | 53.9 |      | 61.9   |     | 54.6 |      |      | 55.8 |     | 53.7 |      | 40.1   |     | 54.4 |
|            | SGR+DAA | **78.0** |     | **68.5** |      | **62.6**   |     | **62.8** |      |      | **56.5** |     | **68.4** |      | **40.8**   |     | **62.6** |

---

> ### Author Response · Authors · 2022-08-09
> **Update**
>
> We have added more analyses and all new experiments to the supplementary materials. The details are as follows:
> - The motivation of the specific formulation of ASP (Eq.4)  is clarified in **Appendix.A**.
> - The comparison between ASP and other multiplicity metric (PMRP) is made in **Appendix.B**.
> - The experiments of more baseline using DAA on different datasets are conducted in **Appendix.C**.
> - The effect of different semantic metrics (CIDEr, BLEU, and Rouge) on DAA is explored in **Appendix.D**.
> - The effect of DAA used in model training with different scale datasets is explored in **Appendix.E**.

---

### Author Response · Authors · 2022-08-02
**[1/2] Common Response**

We appreciate all reviewers for kind reviews and recognizing our work is helpful to image-text matching.

According to reviews, **highlights of this paper include**:

- The proposed evaluation metric and approach are novel and effective. (**CS9A, mhZH, YuA1**)
- The paper is well motivated and well-organized. (**CS9A**)
- It is quite interesting and more reasonable to consider the cross-modal retrieval task as a ranking problem. (**mhZH**)
- The paper will be a valuable contribution to the field of image-text matching. (**mhZH, vghc**)

Both reviewers **CS9A** and **mhZH** mark our work as good soundness, presentation, and contribution. They suggest acceptance.

**Major weakness of this paper is writing issues**, including:
- Some concepts are used in a weird name before definition.
- There are some typos and grammar errors.
- Some notations are confusing.

Based on the above summary, the rebuttal version of the paper has been uploaded.

**The other weakness of this paper is the lack of experiments.**

For some common questions, we conduct more experiments as follows:

## 1. More Baseline ##
Our method is proposed to improve the multiplicity of probabilistic model. Some reviewers suggest testing our method on more non-probabilistic models. Thus, we employ SGRAF [1] as a new baseline, which achieves state-of-the-art performance in non-probabilistic approaches. There are two independent modules in SGRAF (combined by SGR and SAF), we report the experimental results of R@1 and ASP below.

### ***Table 4*** ###

|            |      |     | 1K   | Images |     |      |   |      |     | 5K   | Images |     |      |
|:----------:|:----:|:---:|:----:|:------:|:---:|:----:|:---:|:----:|:---:|:----:|:------:|:---:|:----:|
|            |      | **i2t** |      |        | **t2i** |      |   |      | **i2t** |      |        | **t2i** |      |
| **Method**     | **R@1**  |     | **ASP**  | **R@1**    |     | **ASP**  |   | **R@1**  |     | **ASP**  | **R@1**    |     | **ASP**  |
| SAF*   | 77.1 |     | 54.7 | 62.1   |     | 54.8 |   | 55.1 |     | 54.5 | **40.5**   |     | 54.5 |
| SAF+DAA    | **78.0** |     | **67.2** | **62.8**   |     | **61.6** |   | **56.2** |     | **67.0** | **40.5**   |     | **61.4** |
| SGR*   | 77.2 |     | 53.9 | 61.9   |     | 54.6 |   | 55.8 |     | 53.7 | 40.1   |     | 54.4 |
| SGR+DAA    | **78.0** |     | **68.5** | **62.6**   |     | **62.8** |   | **56.5** |     | **68.4** | **40.8**   |     | **62.6** |
| SGRAF* | 79.4 |     | 54.5 | 63.9   |     | 55.0 |   | 59.0 |     | 54.3 | 42.6   |     | 54.8 |
| SGRAF+DAA  | **80.2** |     | **68.3** | **65.0**   |     | **62.7** |   | **60.0** |     | **68.2** | **43.5**   |     | **62.5** |

"**\***" denotes the model retrained by public codes.

***Ref:***

[1] Diao, et al. "Similarity Reasoning and Filtration for Image-Text Matching." In AAAI 2021.

## 2. Different Semantic Metric ##
CIDEr is one of the most important metrics to evaluate text similarity in image captioning. Thus, we mainly utilize CIDEr as the semantic metric. To validate the effect of different semantic metrics on DAA, we conduct more experiments using CIDEr, BLEU, and Rouge on the SGR model, the results are shown in **Tab.5** as below. Concretely, ASP (C), ASP (B), and ASP (R) are ASP using CIDEr, BLEU, and Rouge as semantic similarity metrics, separately. Meanwhile, we utilize the mean of these three difference matrices to compute ASP (A).

As can be seen, SGR [1] retrained with DAA using CIDEr, BLEU, or Rouge as semantic similarity metrics all can obtain performance improvements on R@1 and ASP. Among them, DAA using CIDEr obtains the most significant improvements on R@1, with the highest ASP (A) score (60.8\%) in text retrieval and the second highest ASP (A) score (65.1\%) in image retrieval. Therefore, compared with BLEU and Rouge, it seems that CIDEr can excavate more precise semantic correspondences.

### ***Table 5*** ###
|           |      |    Image     | Retrieval   |        |         |   |      |   Text      | Retrieval  |         |        |
|:---------:|:----:|:-------:|:---------------:|:-------:|:-------:|:---:|:----:|:-------:|:--------------:|:-------:|:-------:|
| **Method**    | **R@1**  | **ASP(C)** | **ASP(B)**        | **ASP(R)** | **ASP(A)** |   | **R@1**  | **ASP(C)** | **ASP(B)**    | **ASP(R)** | **ASP(A)** |
| SGR (B)   | 77.2 | 53.9    | 53.0            | 52.6    | 53.3    |   | 61.9 | 54.6    | 53.5           | 52.6    | 53.3    |
| B+CIDEr | **78.0** | **68.5**    | 55.7            | 63.3    | 65.1    |   | **62.6** | **62.8**    | 57.9           | 59.1    | **60.8**    |
| B+BLEU  | 77.6 | 55.6    | **68.7**            | 57.8    | 58.7    |   | 62.0 | 59.2    | **58.9**           | 58.2    | 59.6    |
| B+Rouge | 77.4 | 63.2    | 56.6            | **67.1**    | **66.0**    |   | 61.9 | 60.5    | 57.9           | **59.4**    | 60.5    |

***Ref:***

[1] Diao, et al. "Similarity Reasoning and Filtration for Image-Text Matching." In AAAI 2021.

---

### Meta-Review · Area_Chair_t5Dg · 2022-08-21

**Recommendation:** Accept
**Confidence:** Certain

**Metareview:**

This paper investigates the multiple correspondence issue of cross-modal retrieval and proposes a method to improve and evaluate the multiplicity of probabilistic embedding. Sufficient experiments are carried out to prove the effectiveness of the proposed method. In addition, the rebuttal successfully addressed the major concerns and, in the end, there is a general consensus about accepting the paper.

**Award:**

No

---

### Decision · Program_Chairs · 2022-09-14

Accept